# CLASSIFIER-AGNOSTIC SALIENCY MAP EXTRACTION

## ABSTRACT

Extracting saliency maps, which indicate parts of the image important to classification, requires many tricks to achieve satisfactory performance when using classifier-dependent methods. Instead, we propose classifier-agnostic saliency map extraction, which finds all parts of the image that any classifier could use, not just one given in advance. We observe that the proposed approach extracts higher quality saliency maps and outperforms existing weakly-supervised localization techniques, setting the new state of the art result on the ImageNet dataset.

## 1 INTRODUCTION

The success of deep convolutional networks for large-scale object recognition Krizhevsky et al. (2012); Simonyan & Zisserman (2014); Szegedy et al. (2015); He et al. (2016) has spurred interest in utilizing them to automatically detect and localize objects in natural images. Pioneering this direction, Simonyan et al. (2013) and Springenberg et al. (2014) demonstrated that the gradient of the class-specific score of a given classifier could be used for extracting a saliency map of an image. Such classifier-dependent saliency maps can be utilized to analyze the inner workings of a specific network. However, as only the part of the image that is used by a given model is highlighted, these methods are not identifying *all* "evidence" in a given image. They also tend to be noisy, covering many irrelevant pixels and missing many relevant ones. Therefore, much of the recent work has focused on introducing regularization techniques of correcting such classifier-dependent saliency maps. For instance, Selvaraju et al. (2017) propose averaging multiple saliency maps created for perturbed images to obtain a smooth saliency map. We argue, however, that applying tricks and tweaks on top of methods that were designed to analyze inner workings of a given classifier is not a principled way to get saliency maps that focus on *all* useful evidence.

In this work, we aim to find saliency maps indicating pixels which aid classification, i.e. we want to find pixels in the input image such that if they were masked, it would confuse *an unknown classifier*. Assuming we were given a classifier, a naive approximate solution would be to train a generative model to output a mask (a saliency map) confusing that classifier. That can be achieved using a simple GAN-like approach (Goodfellow et al., 2014) where the classifier acts as a fixed discriminator. Unfortunately, as we prove experimentally, this solution suffers from the same issues as prior approaches. We argue that the strong dependence on a given classifier lies at the center of the problem. To tackle this directly we propose to train a saliency mapping that is not strongly coupled with any specific classifier. Our approach, a *class-agnostic saliency map extraction*, can be formulated as a practical algorithm that realizes our goal.

Our focus on classifier-agnostic saliency maps is not our objective per se, it is a remedy that resolves the core problem. The proposed approach results in a neural network based saliency mapping that only depends on an input image. We qualitatively find that it extracts higher quality saliency maps compared to classifier-dependent methods, as can be seen in Fig. 2. Extracted saliency maps show all the evidence without using any symptom-masking methods: difficult to tune regularization penalties (such as total variation), exotic activation functions, complex training procedures or image preprocessing tricks (such as superpixels), etc. We also evaluate our method quantitatively by using the extracted saliency maps for object localization. We observe that the proposed approach outperforms the existing weakly-supervised techniques setting the new state of the art result on the ImageNet dataset and closely approaches the localization performance of a strongly supervised model. Furthermore, we experimentally validate that the proposed approach works reasonably well even for classes unseen during training.

Our method has many potential applications, in which being classifier-agnostic is of primary importance. For instance, in medical image analysis, where we are interested not only in class prediction but also in indicating which part of the image is important to classification. Importantly, however, it is criticial to indicate all parts of the image, which can influence diagnosis, not just ones used by a specific classifier.

## 2 CLASSIFIER-AGNOSTIC SALIENCY MAP EXTRACTION

In this paper, we tackle a problem of extracting a salient region of an input image as a problem of extracting a mapping $m : \mathbb{R}^{W \times H \times 3} \to [0, 1]^{W \times H}$ over an input image $x \in \mathbb{R}^{W \times H \times 3}$. Such a mapping should retain ($= 1$) any pixel of the input image if it aids classification, while it should mask ($= 0$) any other pixel.

### 2.1 CLASSIFIER-DEPENDENT SALIENCY MAP EXTRACTION

Earlier work has largely focused on a setting in which a classifier $f$ was given (Fong & Vedaldi, 2017; Dabkowski & Gal, 2017). These approaches can be implemented as solving the following maximization problem:

$$m = \arg \max_{m'} S(m', f), \tag{1}$$

where $S$ is a score function corresponding to a classification loss. That is,

$$S(m, f) = \frac{1}{N} \sum_{n=1}^{N} \left[ l(f((1 - m(x_n)) \odot x_n), y_n) + R(m(x_n)) \right], \tag{2}$$

where $\odot$ denotes elementwise multiplication (masking), $R(m)$ is a regularization term and $l$ is a classification loss, such as cross-entropy. We are given a training set $D = \{(x_1, y_1), \ldots, (x_N, y_N)\}$. This optimization procedure could be interpreted as finding a mapping $m$ that maximally confuses a given classifier $f$. We refer to it as a *classifier-dependent saliency map extraction*.

A mapping $m$ obtained with a classifier $f$ may differ from a mapping $m'$ found using $f'$, even if both classifiers are equally good in respect to a classification loss for both original and masked images, i.e. $L(0, f) = L(0, f')$ and $L(m, f) = L(m', f')$, where

$$L(m, f) = \frac{1}{N} \sum_{n=1}^{N} l(f((1 - m(x_n)) \odot x_n), y_n). \tag{3}$$

This property is against our definition of the mapping $m$ above, which stated that any pixel which helps classification should be indicated by the mask (a saliency map) with 1. The reason why this is possible is these two equally good, but distinct classifiers may use different subsets of input pixels to perform classification.

**An example** This behaviour can be intuitively explained with a simple example, illustrating an extreme special case. Let us consider a data set in which all instances consist of two identical copies of images concatenated together, that is, for all $x_n$, $x_n = [x'_n; x'_n]$, where $x'_n \in \mathbb{R}^{W/2 \times H \times 3}$. For such a data set, there exist at least two classifiers, $f$ and $f'$, with the same classification loss. The classifier $f$ uses only the left half of the image, while $f'$ uses the other half. Each of the corresponding mappings, $m$ and $m'$, would then indicate a region of interest only on the corresponding half of the image. When the input image does not consist of two concatenated copies of the same image, it is unlikely that two equally good classifiers will use disjoint sets of input pixels. Our example is to show an extreme case when it is possible.

### 2.2 CLASSIFIER-AGNOSTIC SALIENCY MAP EXTRACTION

In order to address the issue of saliency mapping's dependence on a single classifier, we propose to alter the objective function in Eq. (1) to consider not only a single fixed classifier but all possible classifiers weighted by their posterior probabilities. That is,

$$m = \arg \max_{m'} \mathbb{E}_f \left[ S(m', f) \right], \tag{4}$$

where the posterior probability, $p(f|D, m')$, is defined to be proportional to the exponentiated classification loss $L$, i.e., $p(f|D, m') \propto p(f) \exp(-L(m', f))$. Solving this optimization problem is equivalent to searching over the space of all possible classifiers, and finding a mapping $m$ that works with all of them. As we parameterize $f$ as a convolutional network (with parameters denoted as $\theta_f$), the space of all possible classifiers is isomorphic to the space of its parameters. The proposed approach considers all the classifiers and we call it a *classifier-agnostic saliency map extraction*.

In the case of the simple example above, where each image contains two copies of a smaller image, both $f$ and $f'$, which respectively look at one and the other half of an image, the posterior probabilities of these two classifiers would be the same[1]. Solving Eq. (4) implies that a mapping $m$ must minimize the loss $S$ for both of these classifiers.

## 2.3 ALGORITHM

The optimization problem in Eq. (4) is, unfortunately, generally intractable. This arises from the intractable expectation over the posterior distribution. Furthermore, the expectation is inside the optimization loop for the mapping $m$, making it even harder to solve.

Thus, we approximately solve this problem by simultaneously estimating the mapping $m$ and the expected objective. First, we sample one $f^{(k)}$ with the posterior probability $p(f|D, m^{(k-1)})$ by taking a single step of stochastic gradient descent (SGD) on the classification loss with respect to $\theta_f$ (classifier $f$ parameters) with a small step size:

---

**Algorithm 1:** Classifier-agnostic saliency map extraction

**input** : an initial classifier $f^{(0)}$,
    an initial mapping $m^{(0)}$,
    dataset $D$,
    number of iterations $K$
**output** : the final mapping $m^{(K)}$

Initialize a sample set $F^{(0)} = \left\{ f^{(0)} \right\}$.

**for** $k \leftarrow 1$ **to** $K$ **do**
    $\theta_{f^{(k)}} \leftarrow \theta_{f^{(k-1)}} - \eta_f \nabla_{\theta_f} L(m^{(k-1)}, f^{(k-1)})$
    $F^{(k)} \leftarrow F^{(k-1)} \cup \left\{ f^{(k)} \right\}$
    $f' \leftarrow \text{Sample}(F^{(k)})$
    $\theta_{m^{(k)}} \leftarrow \theta_{m^{(k-1)}} + \eta_m \nabla_{\theta_m} S(m^{(k-1)}, f')$
    $F^{(k)} \leftarrow \text{Thin}(F^{(k)})$

---

$$\theta_{f^{(k)}} \leftarrow \theta_{f^{(k-1)}} - \eta_f \nabla_{\theta_f} L(m^{(k-1)}, f^{(k-1)}). \tag{5}$$

This is motivated by earlier work (Welling & Teh, 2011; Mandt et al., 2017) which showed that SGD performs approximate Bayesian posterior inference.

We have up to $k + 1$ samples[2] from the posterior distribution $F^{(k)} = \left\{ f^{(0)}, \ldots, f^{(k)} \right\}$. We sample[3] $f' \in F^{(k)}$ to get a single-sample estimate of $m$ in in Eq. (4) by computing $S(m^{(k-1)}, f')$. Then, we use it to obtain an updated $\theta_m$ (mapping $m$ parameters) by

$$\theta_{m^{(k)}} \leftarrow \theta_{m^{(k-1)}} + \eta_m \nabla_{\theta_m} S(m^{(k-1)}, f'). \tag{6}$$

We alternate between these two steps until $m^{(k)}$ converges (cf. Alg. 1). Note, that our algorithm resembles the training procedure of GANs (Goodfellow et al., 2014), where mapping $m$ takes the role of a generator and the classifier $f$ can be understood as a discriminator. Fan et al. (2017) also applied adversarial training in order to achieve better saliency maps. The relation to the work is discussed in details in Section 6.

**Score function** The score function $S(m, f)$ estimates the quality of the saliency map extracted by $m$ given a data set and a classifier $f$. The score function must be designed to balance the precision and recall. The precision refers to the fraction of relevant pixels among those marked by $m$ as relevant, while the recall is the fraction of pixels correctly marked by $m$ as relevant among all the relevant pixels. In order to balance these two, the score function often consists of two terms.

The first term is aiming to ensure that all relevant pixels are included (high recall). As in Eq. (2), a popular choice has been the classification loss based on an input image $x$ masked out by $1 - m(x)$. In our preliminary experiments, however, we noticed that this approach leads to obtaining masks with adversarial artifacts. We hence propose to use the entropy $\mathcal{H}(f((1 - m(x)) \odot x))$ instead. This makes generated masks cover all salient pixels in the input, avoiding masks that may sway the class

---

[1]We assume a flat prior, i.e., $p(f) = c$.

[2]A usual practice of "thinning" may be applied, leading to a fewer than $k$ samples.

[3]We set the chance of selecting $f^{(k)}$ to 50% and we spread the remaining 50% uniformly over $F^{(k-1)}$.

prediction to a different, but semantically close class. For example, from one dog species to another. The second term, $R(m)$, excludes a trivial solution, $m$ simply outputting an all-ones saliency map, which would achieve maximal recall with low very precision. In order that, we must introduce a regularization term. Some of the popular choices include total variation (Rudin et al., 1992) and $L^1$ norm. For simplicity, we use the latter only.

In summary, we use the following score function for the class-agnostic saliency map extraction:

$$S(m, f) = \frac{1}{N} \sum_{n=1}^{N} \Big[ \mathcal{H}\big(f((1 - m(x_n)) \odot x_n)\big) - \lambda_R \|m(x_n)\|_1 \Big], \tag{7}$$

where $\lambda_R > 0$ is a regularization coefficient.

**Thinning**   As the algorithm collects a set of classifiers, $f^{(k)}$'s, from the posterior distribution, we need a strategy to keep a small subset of them. An obvious approach would be to keep all classifiers but this does not scale well with the number of iterations. We propose and empirically evaluate a few strategies. The first three of them assume a fixed size of $F^{(k)}$. Namely, keeping the first classifier only, denoted by **F** ($F^{(k)} = \{f^{(0)}\}$), the last only, denoted by **L** ($F^{(k)} = \{f^{(k)}\}$) and the first and last only, denoted by **FL** ($F^{(k)} = \{f^{(0)}, f^{(k)}\}$). As an alternative, we also considered a growing set of classifiers where we only keep one every 1000 iterations (denoted by **L1000**) but whenever $|F^{(k)}| = 30$, we randomly remove one from the set. Analogously, we experimented with **L100**.

**Classification loss**   Although we described our approach using the classification loss computed only on masked images, as in Eq. (3), it is not necessary to define the classification loss exactly in this way. In the preliminary experiments, we noticed that the following alternative formulation, inspired by adversarial training (Szegedy et al., 2013), works better:

$$L(m, f) = \frac{1}{2N} \sum_{n=1}^{N} \left[ l(f((1 - m(x_n)) \odot x_n), y_n) + l(f(x_n), y_n) \right]. \tag{8}$$

We thus use the loss as defined above in the experiments. We conjecture that it is advantageous over the original one in Eq. (3), as the additional term prevents the degradation of the classifier's performance on the original, unmasked images while the first term encourages the classifier to collect new pieces of evidence from the images that are masked.

## 3 EXPERIMENTAL SETTINGS

**Dataset: ImageNet**   Our models were trained on the official ImageNet training set with ground truth class labels Deng et al. (2009). We evaluate them on the validation set. Depending on the experiment, we use ground truth class or localization labels.

**Reproducibility**   We made our code publicly available at `MASKED`.

### 3.1 ARCHITECTURES

**Classifier $f$ and mapping $m$**   We use ResNet-50 (He et al., 2016) as a classifier $f$ in our experiments. We follow an encoder-decoder architecture for constructing a mapping $m$. The encoder is implemented also as a ResNet-50 so its weights can be shared with the classifier or it can be separate. We experimentally find that sharing is beneficial. The decoder is a deep deconvolutional network that ultimately outputs the mask of an input image. The overall architecture is shown in Fig. 1. Details of the architecture and training procedure are in the appendix.

**Regularization coefficient $\lambda_R$**   As noticed by Fan et al. (2017), it is not trivial to find an optimal regularization coefficient $\lambda_R$. They proposed an adaptive strategy which gets rid of the manual selection of $\lambda_R$. We, however, find it undesirable due to the lack of control on the average size of the saliency map. Instead, we propose to control the average number of relevant pixels by manually setting $\lambda_R$, while applying the regularization term $R(m(x))$ only when there is a disagreement between $f(x)$ and $f((1 - m(x)) \odot x)$. We then set $\lambda_R$ for each experiment such that approximately 50% of pixels in each image are indicated as relevant by a mapping $m$. In the preliminary experiments, we further noticed that this approach avoids problematic behavior when an image contains small objects, earlier observed by Fong & Vedaldi (2017).

We also noticed that the training of mapping $m$ is more stable and effective when we use only images that the classifier is not trivially confused on, i.e. predicts the correct class for the original images.

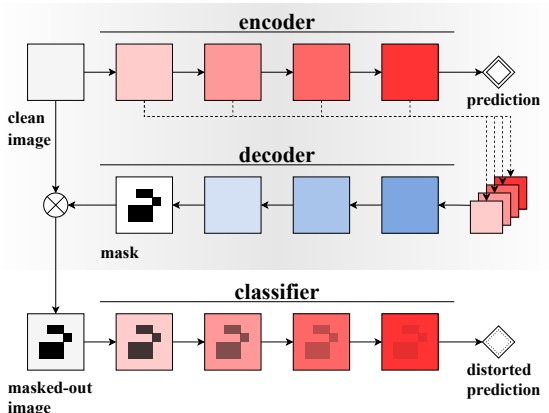

### 3.2 EVALUATION

In our experiments we only use a single architecture explained in subsection 3.1. We use the abbreviation CASM (classifier-agnostic saliency mapping) to denote the final model obtained using the proposed method. Our baseline model (Baseline) is of the same architecture and it is trained with a fixed classifier (classifier-dependent saliency mapping) realized by following thinning strategy **F**.

Figure 1: The overall architecture. The mapping $m$ consists of an encoder and an decoder and is shown at the top with gray background. The additional forward pass (when classifier acts on masked-out image) is needed during training only.

Following the previous work (Cao et al., 2015; Fong & Vedaldi, 2017; Zhang et al., 2016) we discretize our mask by computing

$$b_{ij}(x) = \begin{cases} 1, & \text{if } m_{ij}(x) \geq \alpha \overline{m}(x) \\ 0, & \text{otherwise} \end{cases},$$

where $\overline{m}(x)$ is the average mask intensity and $\alpha$ is a hyperparameter. We simply set $\alpha$ to 1, hence the average of pixel intensities is the same for the input mask $m(x)$ and the discretized *binary mask* $b(x)$. To focus on the most dominant object we take the largest connected component of the binary mask to obtain the *binary connected mask*.

**Visualization** We visualize the learned mapping $m$ by inspecting the saliency map of each image in three different ways. First, we visualize the **masked-in image** $b(x) \odot x$, which ideally leaves only the relevant pixels visible. Second, we visualize the **masked-out image** $(1 - b(x)) \odot x$, which highlights pixels irrelevant to classification. Third, we visualize the **inpainted masked-out image** using an inpainting algorithm (Telea, 2004). This allows us to inspect whether the object that should be masked out cannot be easily reconstructed from nearby pixels.

**Classification by multiple classifiers** In order to verify our claim that the proposed approach results in a classifier-agnostic saliency mapping, we evaluate a set of classifers[4] on the validation sets of masked-in images, masked-out images and inpainted masked-out images. If our claim is correct, we expect the inpainted masked-out images created by our method to break these classifiers, while the masked-in images would suffer minimal performance degradation.

**Object localization** As the saliency map can be used to find the most dominant object in an image, we can evaluate our approach on the task of weakly supervised localization. To do so, we use the ILSVRC'14 localization task. We compute the bounding box of an object as the tightest box that covers the binary connected mask.

We use three metrics to quantify the quality of localization. First, we use the official metric (**OM**) from the ImageNet localization challenge, which considers the localization successful if at least one ground truth bounding box has IOU with predicted bounding box higher than 0.5 *and* the class prediction is correct. Since **OM** is dependent on the classifier, from which we have sought to make our mapping independent, we use another widely used metric, called localization error (**LE**), which only depends on the bounding box prediction Cao et al. (2015); Fong & Vedaldi (2017). Lastly, we evaluate the original saliency map, of which each mask pixel is a continuous value between 0 and 1, by the continuous **F1** score. Precision $P$ and recall $R$ are defined as the following:

$$P = \frac{\sum_{(i,j) \in B^*(x)} m_{ij}(x)}{\sum_{ij} m_{ij}(x)} \text{ and } R = \frac{\sum_{(i,j) \in B^*(x)} m_{ij}(x)}{|B^*(x)|},$$

---

[4]We train twenty ResNet-50 models with different initial random sets of parameters in addition to the classifiers from torchvision.models (https://pytorch.org/docs/master/torchvision/models.html): densenet121, densenet169, densenet201, densenet161, resnet18, resnet34, resnet50, resnet101, resnet152, vgg11, vgg11_bn, vgg13, vgg13_bn, vgg16, vgg16_bn, vgg19 and vgg19_bn.

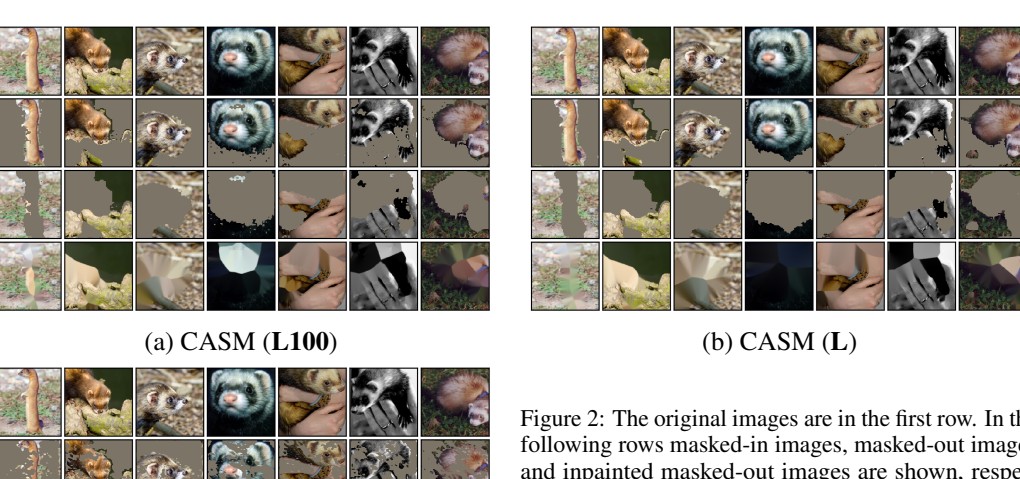

(a) CASM (**L100**)              (b) CASM (**L**)

Figure 2: The original images are in the first row. In the following rows masked-in images, masked-out images and inpainted masked-out images are shown, respectively. Note that the proposed approach (a-b) remove all relevant pixels and hence the inpainted images show the background only. Seven randomly selected consecutive images from validation set are presented here. Please look into the appendix for extra visualizations.

(c) Baseline

where $B^*(x)$ is the ground truth bounding box. We compute F1 scores against all the ground truth bounding boxes for each image and report the highest one among them as its final score.

# 4 RESULTS AND ANALYSIS

**Visualization and statistics** We randomly select seven consecutive images from the validation set and input them to two instances of CASM (each using a different thinning strategy – **L** or **L100**) and Baseline. We visualize the original (clean), masked-in, masked-out and inpainted masked-out images in Fig. 2. The proposed approach produces clearly better saliency maps, while the classifier-dependent approach (Baseline) produces so-called adversarial masks (Dabkowski & Gal, 2017).

We further compute some statistics of the saliency maps generated by CASM and Baseline over the validation set. The masks extracted by CASM exhibit lower total variation ($2.5 \times 10^3$ vs. $7.0 \times 10^3$), indicating that CASM produced more regular masks, despite the lack of explicit TV regularization. The entropy of mask pixel intensities is much smaller for CASM ($0.05$ vs. $0.21$), indicating that the mask intensities are closer to either $0$ or $1$ on average. Furthermore, the standard deviation of the masked out volume is larger with CASM ($0.19$ vs. $0.14$), indicating that CASM is capable of producing saliency maps of varying sizes dependent on the input images.

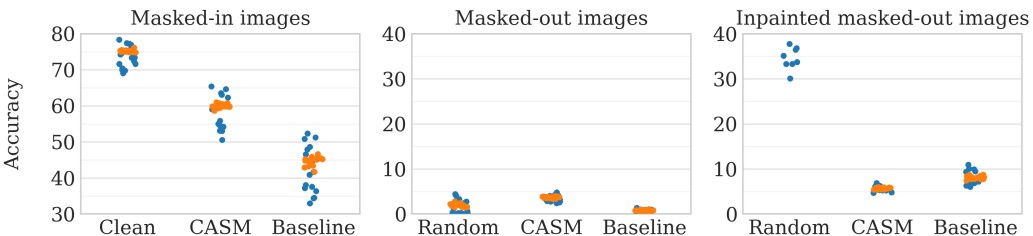

Figure 3: The classification accuracy of the ImageNet-trained convolutional networks on masked-in images (left), masked-out images (center) and inpainted masked-out images (right). Orange and blue dots correspond to ResNet-50 models and all the other types of convolutional networks, respectively. We observe that the inpainted masked-out images obtained using Baseline are easier to classify than those using CASM, because Baseline fails to mask out all the relevant pixels, unlike CASM. On the right panel, most of the classifiers evaluated on images with random masks achieve accuracy higher than 40% and are not shown. We add jitters in the x-axis to make each dot more distinct from the others visibly.

Table 1: Localization evaluation using **OM** and **LE** scores. We report the better accuracy between those reported in the original papers or by Fong & Vedaldi (2017).

| Model | OM↓ | LE↓ |
|---|---|---|
| *Our:* | | |
| Baseline | 62.7 | 53.5 |
| CASM | **48.6** | **36.1** |
| Fan et al. (2017) | 54.5 | 43.5 |
| *Weakly supervised:* | | |
| Zeiler & Fergus (2014) | - | 48.6 |
| Zhou et al. (2016) | 56.4 | 48.1 |
| Selvaraju et al. (2017) | - | 47.5 |
| Fong & Vedaldi (2017) | - | 43.1 |
| Mahendran & Vedaldi (2016) | - | 42.0 |
| Simonyan et al. (2013) | - | 41.7 |
| Cao et al. (2015) | - | 38.8 |
| Zhang et al. (2016) | - | 38.7 |
| Dabkowski & Gal (2017) | - | 36.7 |
| *Supervised:* | | |
| Simonyan & Zisserman (2014) | - | **34.3** |

Table 2: Ablation study. **S** refers to the choice of a score function (E: entropy, C: classification loss), **Shr** to whether the encoder and classifier are shared (Y: yes, N: no) and **Thin** to the thinning strategies.

| | S | Shr | Thin | OM↓ | LE↓ | F1↑ |
|---|---|---|---|---|---|---|
| (a) | E | Y | F | 62.7 | 53.5 | 49.0 |
| (b) | E | Y | L | 49.0 | 36.5 | 61.7 |
| (c) | E | Y | FL | 52.7 | 41.3 | 57.2 |
| (d) | E | Y | L1000 | 48.7 | 36.2 | 61.6 |
| (e) | E | Y | L100 | **48.6** | **36.1** | 61.4 |
| (f) | C | Y | F | 80.8 | 75.9 | 42.5 |
| (g) | C | Y | L | 49.5 | 37.0 | **62.2** |
| (h) | C | Y | L100 | 49.7 | 37.3 | 62.1 |
| (i) | E | N | F | - | 55.5 | - |
| (j) | E | N | L | - | 47.2 | - |
| (k) | E | N | L100 | - | 46.8 | - |

**Classification**    As shown on the left panel of Figure 3, the entire set of classifiers suffers less from the masked-in images produced by CASM than those by Baseline. We, however, notice that most of the classifiers fail to classify the masked-out images produced by Baseline, which we conjecture is due to the adversarial nature of the saliency maps produced by Baseline approach. This is confirmed by the right panel which shows that simple inpainting of the masked-out images dramatically increases the accuracy when the saliency maps were produced by Baseline. The inpainted masked-out images by CASM, on the other hand, do not benefit from inpainting, because it truly does not maintain any useful evidence for classification.

**Localization**    We report the localization performance of CASM, Baseline and prior works in Table 1 using two different metrics. Most of the existing approaches, except for Fan et al. (2017), assume the knowledge of the target class, unlike our work. CASM performs better than all prior approaches including the classifier-dependent Baseline. The difference is statistically significant. For ten separate training runs with random initialization the worst scores 36.3 and the best 36.0 with the average of 36.1. The fully supervised approach is the only approach that outperforms CASM.

**Thinning strategies**    In Table 2 (a–e), we compare the five thinning strategies described earlier, where **F** is equivalent to the Baseline. According to **LE** and **OM** metrics, the strategies **L100** and **L1000** perform better than the others, closely followed by **L**. These three strategies also perform the best in term of **F1**.

**Sharing the encoder and classifier**    As clear from Fig. 1, it is not necessary to share the parameters of the encoder and classifier. Our experiments, however, reveal that it is always beneficial to share them as shown in Table 2.

**Score function**    Unlike Fan et al. (2017), we use separate score functions for training the classifier and the saliency mapping. We empirically observe in Table 2 that the proposed use of entropy as a score function results in a better mapping in term of **OM** and **LE**. The gap, however, narrows as we use better thinning strategies. On the other hand, the classification loss is better for **F1** as it makes CASM focus on the dominant object only. Because we take the highest score for each ground truth bounding box, concentrating on the dominant object yields higher scores.

## 5    UNSEEN CLASSES

Since the proposed approach does not require knowing the class of the object to be localized, we can use it with images that contain objects of classes that were not seen during training neither by the classifier $f^{(k)}$ nor the mapping $m$. We explicitly test this capability by training five different CASMs on five subsets of the original training set of ImageNet.

Table 3: Localization errors (**LE** in %, ↓) of the models trained on a subset of classes. Each row corresponds to the training subset of classes and each column to the test subset of classes. Error rates on the previously unseen classes are marked with gray shade.

|             | A    | B    | C    | D    | E    | F    | All  |
|------------:|------|------|------|------|------|------|------|
| F           | 46.5 | 46.4 | 48.1 | 45.0 | 45.7 | 41.3 | 44.9 |
| E, F        | 39.5 | 41.2 | 43.1 | 40.3 | 39.5 | 38.7 | 40.0 |
| D, E, F     | 37.9 | 39.3 | 40.0 | 38.0 | 38.0 | 37.4 | 38.1 |
| C, D, E, F  | 38.2 | 38.5 | 39.9 | 37.9 | 37.9 | 37.8 | 38.1 |
| B, C, D, E, F | 36.7 | 36.8 | 39.9 | 37.4 | 37.0 | 37.0 | 37.4 |
| -           | 35.6 | 36.1 | 39.0 | 37.0 | 36.6 | 36.7 | 36.9 |

We first divide the 1000 classes into five disjoint subsets (denoted as A, B, C, D, E and F) of sizes 50, 50, 100, 300, 300 and 200, respectively. We train our models (in all stages) on 95% images (classes in B, C, D, E and F), 90% images (classes in C, D, E and F), 80% images (classes in D, E and F), 50% images (classes in E and F) and finally on 20% of images only (classes in F only). Then, we test each saliency mapping on all the six subsets of classes independently. We use the thinning strategy **L** for computational efficiency in each case.

All models generalize well and the difference between their accuracy on seen or unseen classes is negligible (with exemption of the model trained on 20% of classes). The general performance is a little poorer which can be explained by the smaller training set. In Table 3, we see that the proposed approach works well even for localizing objects from previously unseen classes. The gap in the localization error between the seen and unseen classes grows as the training set shrinks. However, with a reasonably sized training set, the difference between the seen and unseen classes is small. This is an encouraging sign for the proposed model as a *class-agnostic saliency map*.

# 6 RELATED WORK

The adversarial localization network Fan et al. (2017) is perhaps the most closely related to our work. Similarly to ours, they simultaneously train the classifier and the saliency mapping which does not require the object's class at test time. There are four major differences between that work and ours. First, we use the entropy as a score function for training the mapping, whereas they used the classification loss. This results in obtaining better saliency maps as we have shown earlier. Second, we make the training procedure faster thanks to tying the weights of the encoder and the classifier, which also results in a much better performance. Third, we do not let the classifier shift to the distribution of masked-out images by continuing training it on both clean and masked-out images. Finally, their mapping relies on superpixels to build more contiguous masks which may miss small details due to inaccurate segmentation and makes the entire procedure more complex. Our approach solely works on raw pixels without requiring any extra tricks.

Dabkowski & Gal (2017) also train a separate neural network dedicated to predicting saliency maps. However, their approach is a classifier-dependent method and, as such, a lot of effort is devoted to preventing generating adversarial masks. Furthermore, the authors use a complex training objective with multiple hyperparameters which also has to be tuned carefully. On a final note, their model needs a ground truth class label which limits its use in practice.

# 7 CONCLUSIONS

In this paper, we proposed a new framework for classifier-agnostic saliency map extraction which aims at finding a saliency mapping that works for all possible classifiers weighted by their posterior probabilities. We designed a practical algorithm that amounts to simultaneously training a classifier and a saliency mapping using stochastic gradient descent. We qualitatively observed that the proposed approach extracts saliency maps that cover all the relevant pixels in an image and that the masked-out images cannot be easily recovered by inpainting, unlike for classifier-dependent approaches. We further observed that the proposed saliency map extraction procedure outperforms all existing weakly supervised approaches to object localization and can also be used on images containing objects from previously unseen classes, paving a way toward class-agnostic saliency map extraction.

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

APPENDIX

ARCHITECTURE AND TRAINING PROCEDURE

**Classifier $f$ and mapping $m$**   As mentioned before, we use ResNet-50 (He et al., 2016) as a classifier $f$ in our experiments. We follow the encoder-decoder architecture for constructing a mapping $m$. The encoder is implemented also as a ResNet-50 so its weights can be shared with the classifier or it can be separate. We experimentally find that sharing is beneficial. The decoder is a deep deconvolutional network that ultimately outputs the mask of an input image. The input to the decoder consists of all hidden layers of the encoder which are directly followed by a downscaling operation. We upsample them to be of the same size and concatenate them into a single feature map $H$. This upsampling operation is implemented by first applying $1{\times}1$ convolution with 64 filters, followed by batch normalization, ReLU non-linearity and then rescaling to $56{\times}56$ pixels (using bilinear interpolation). Finally, a single $3{\times}3$ convolutional filter followed by sigmoid activation is applied on $H$ and the output is upscaled to a $224{\times}224$ pixel-sized mask using proximal interpolation. The overall architecture is shown in Fig. 1.

**Training procedure**   We initialize the classifier $f^{(0)}$ by training it on the entire training set. We find this pretraining strategy facilitates learning, particularly in the early stage. In practice we use the pretrained ResNet-50 from torchvision model zoo. We use vanilla SGD with a small learning rate of 0.001 (with momentum coefficient set to 0.9 and weight-decay coefficient set to $10^{-4}$) to continue training the classifier with the mixed classification loss as in Eq. (8). To train the mapping $m$ we use Adam (Kingma & Ba, 2014) with the learning rate $l_0 = 0.001$ (with weight-decay coefficient set to $10^{-4}$) and all the other hyperparameters set to default values. We fix the number of training epochs to 70 (each epoch covers only a random 20% of the training set).

RESIZING

We noticed that the details of the resizing policy preceding the evaluation procedures **OM** and **LE** vary between different works. The one thing they have in common is that the resized image is always $224{\times}224$ pixels. The two main approaches are the following.

- The image in the original size is resized such that the smaller edge of the resulting image is 224 pixels long. Then, the central $224{\times}224$ crop is taken. The original aspect ratio of the objects in the image is preserved. Unfortunately, this method has a flaw – it may be impossible to obtain IOU > 0.5 between predicted localization box and the ground truth box when than a half of the bounding box is not seen by the model.
- The image in the original size is resized directly to $224{\times}224$ pixels. The advantage of this method is that the image is not cropped and it is always possible to obtain IOU > 0.5 between predicted localization box and the ground truth box. However, the original aspect ratio is distorted.

The difference in **LE** scores for different resizing strategies should not be large. For CASM it is 0.6%. In this paper, for CASM, we report results for the first method.

VISUALIZATIONS

In the remained of the appendix we replicate the content of Fig. 2 for sixteen randomly chosen classes. That is, in each figure we visualize saliency maps obtained for seven consecutive images from the validation set. The original images are in the first row. In the following rows masked-in images, masked-out images and inpainted masked-out images are shown. As before, we used two instances of CASM (each using a different thinning strategy – **L** or **L100**) and Baseline.

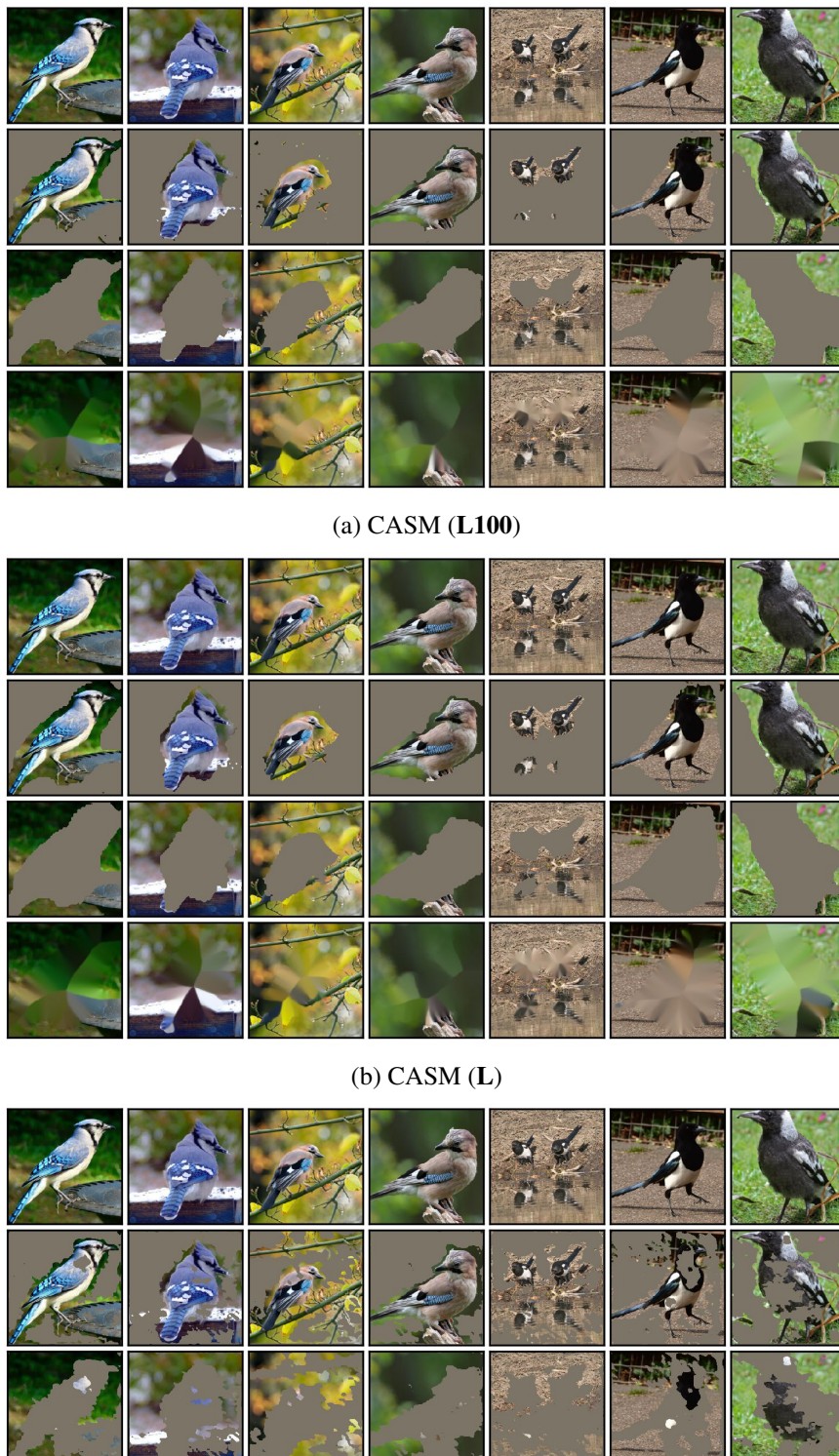

(a) CASM (**L100**)

(b) CASM (**L**)

(c) Baseline

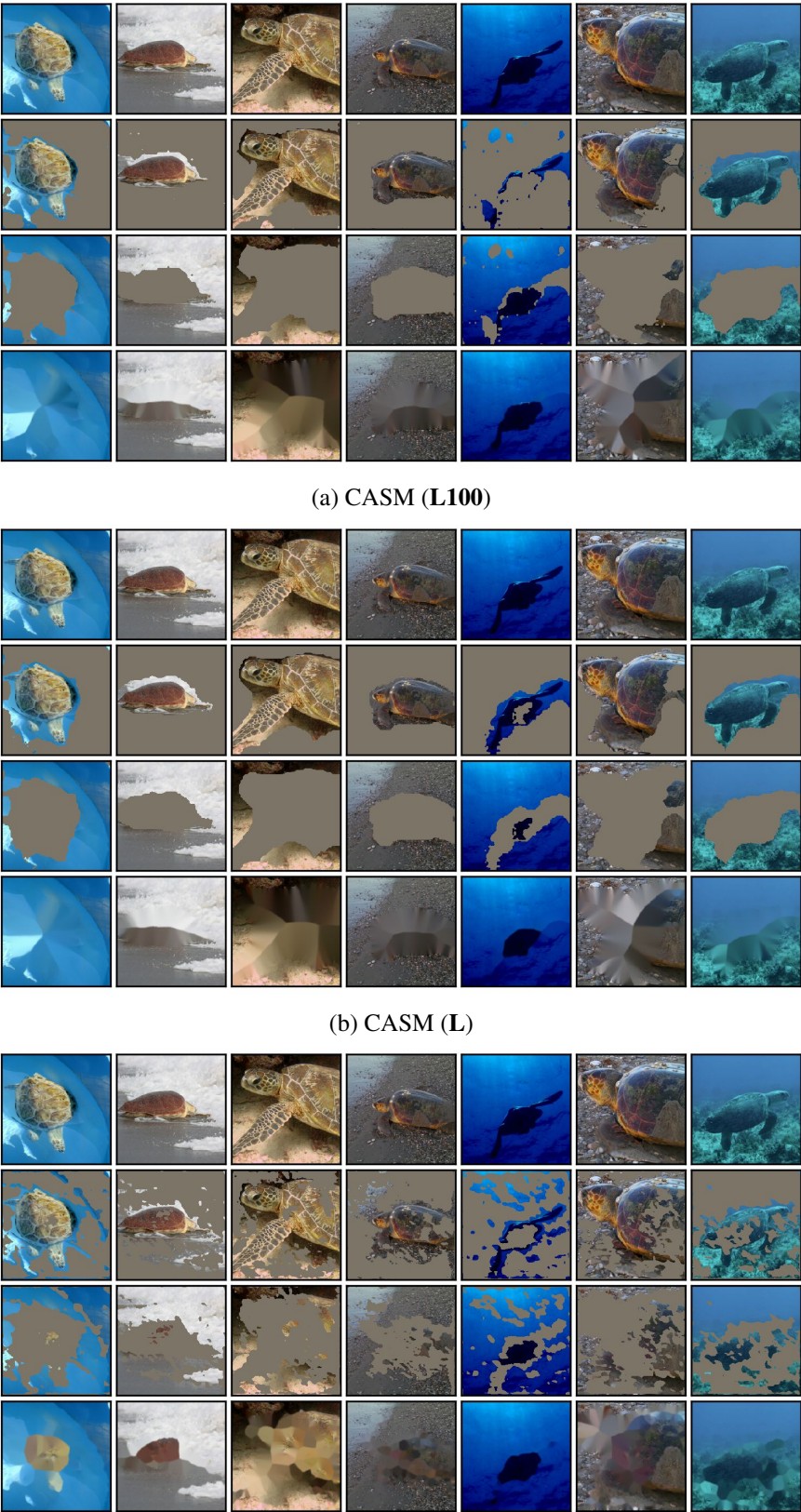

(a) CASM (**L100**)

(b) CASM (**L**)

(c) Baseline

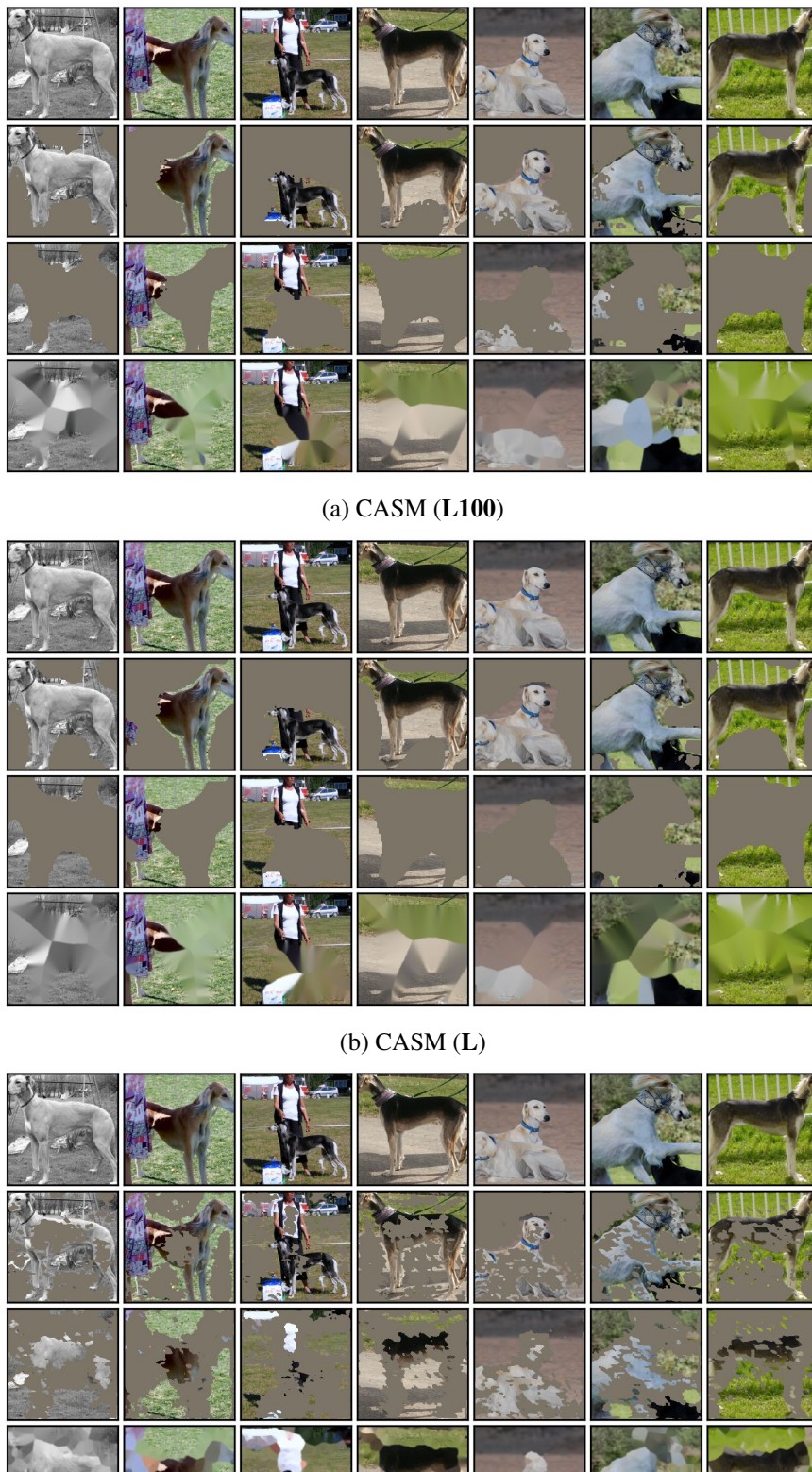

(a) CASM (**L100**)

(b) CASM (**L**)

(c) Baseline

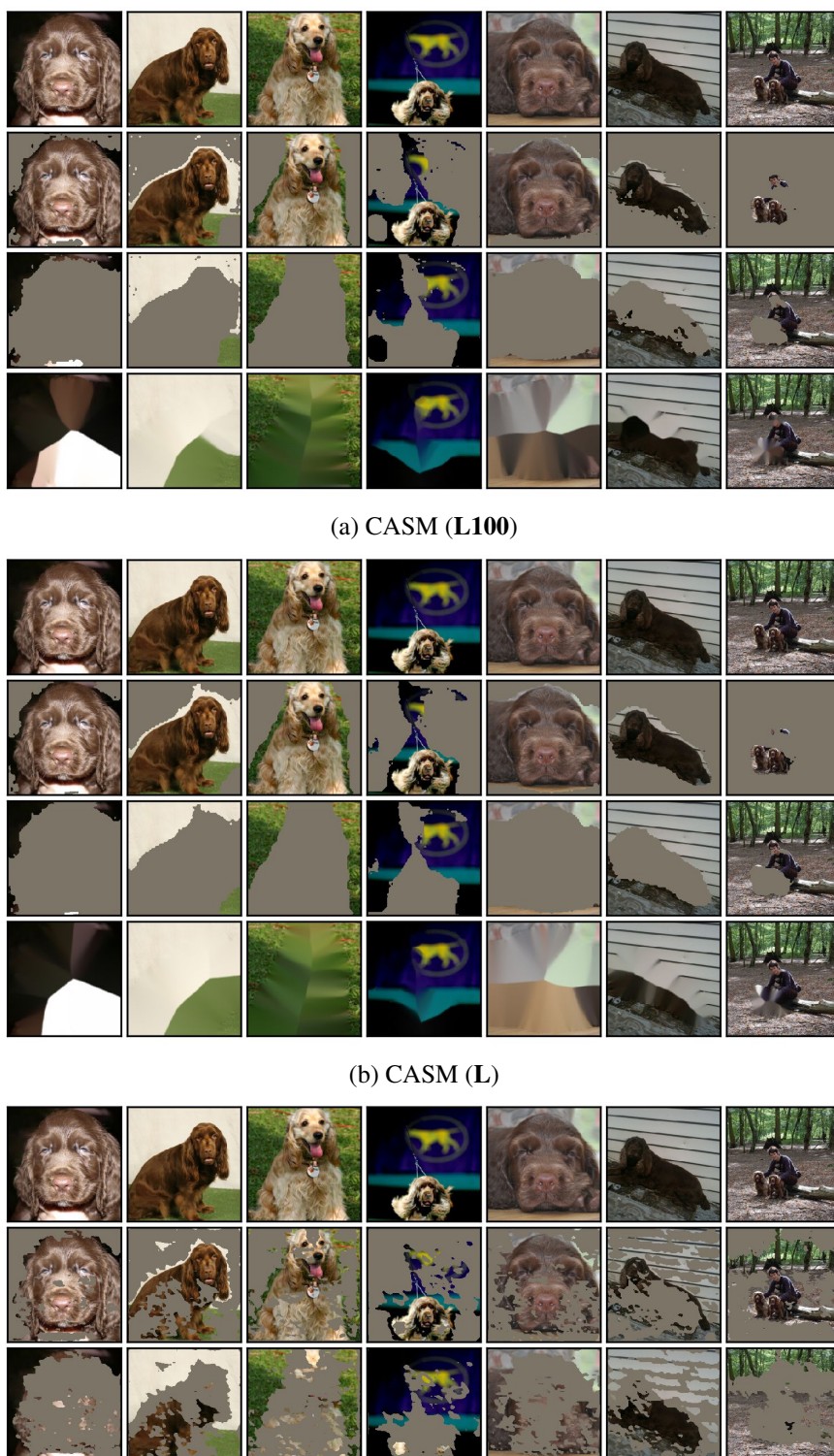

(a) CASM (**L100**)

(b) CASM (**L**)

(c) Baseline

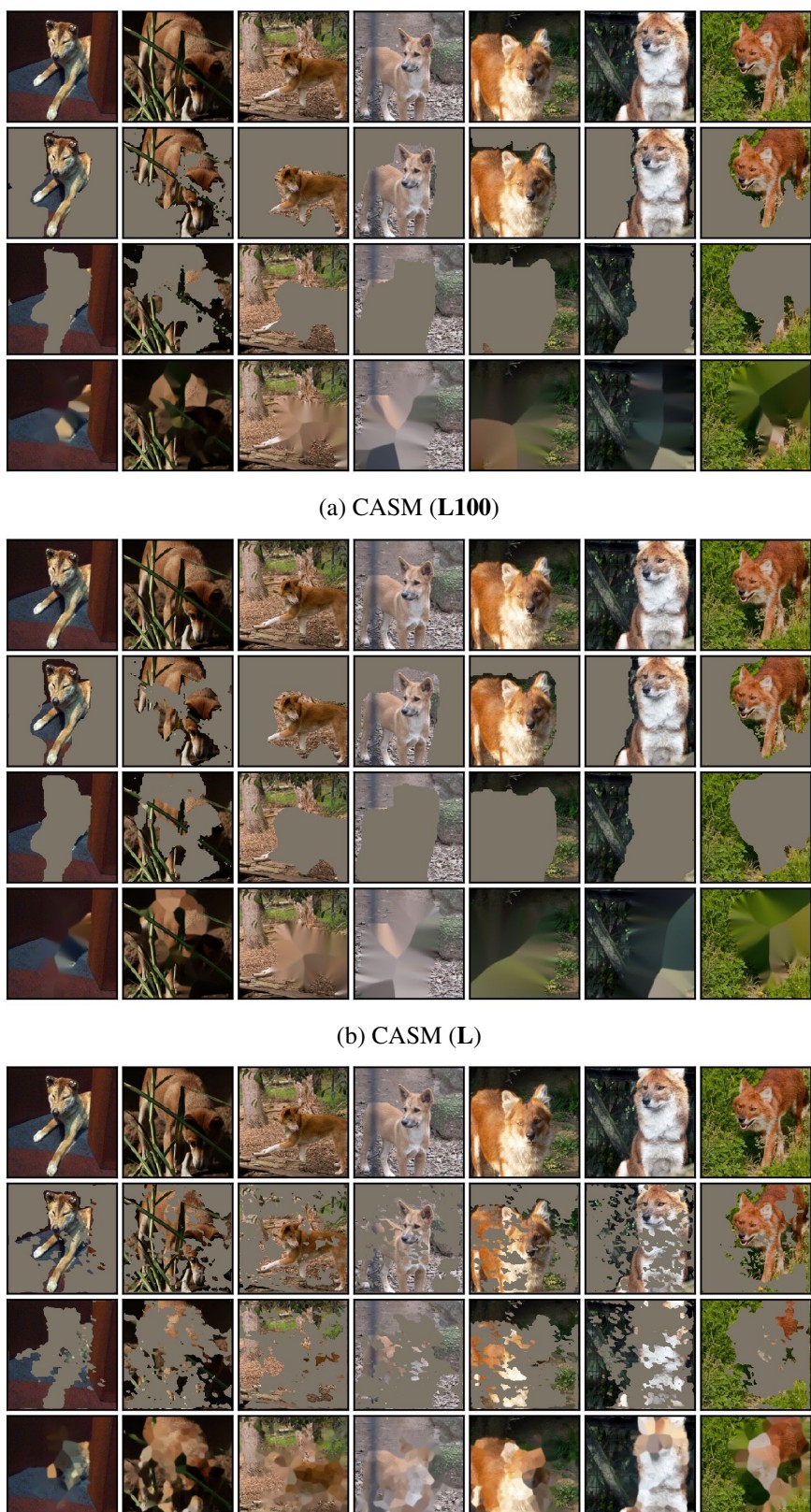

(a) CASM (**L100**)

(b) CASM (**L**)

(c) Baseline

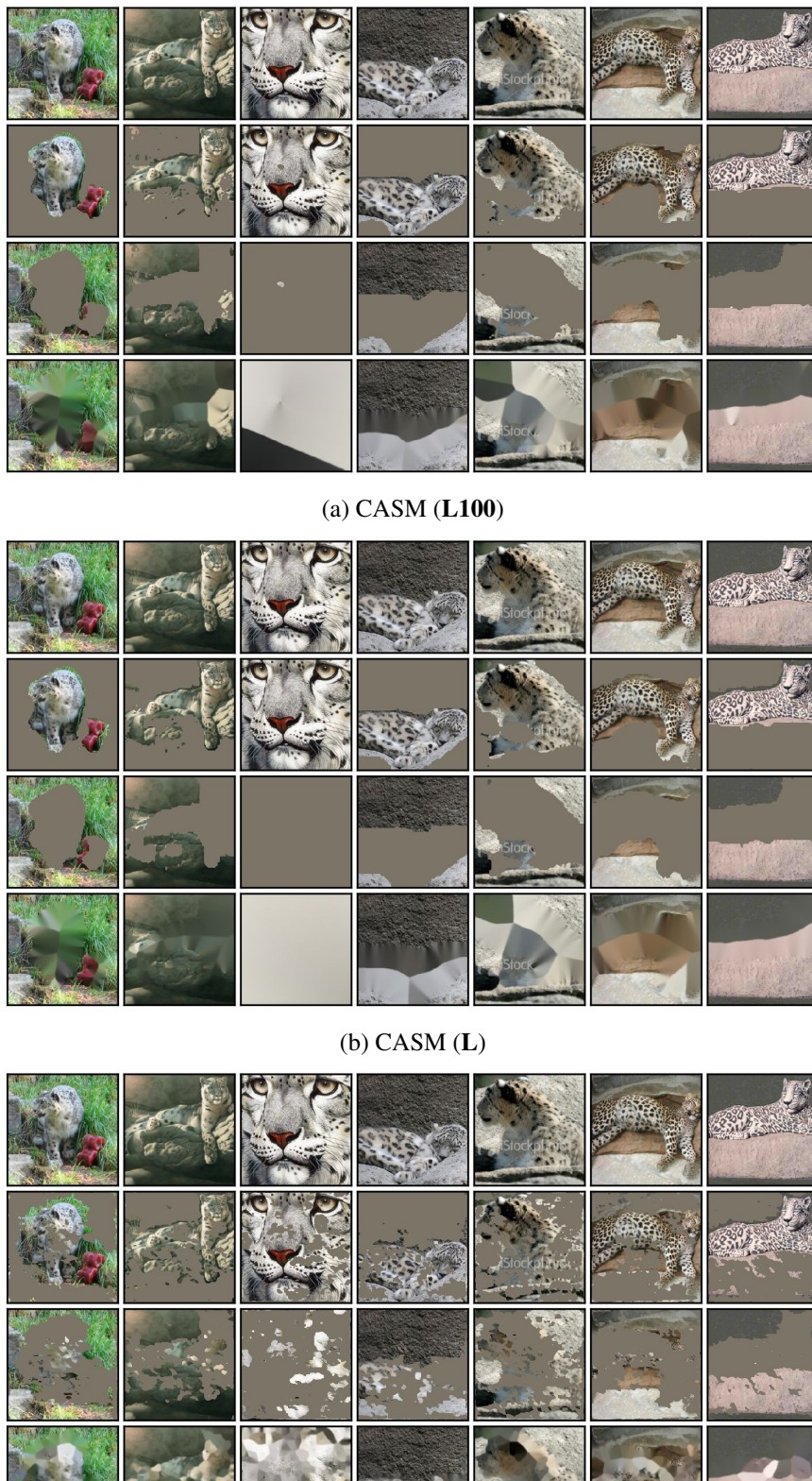

(a) CASM (**L100**)

(b) CASM (**L**)

(c) Baseline

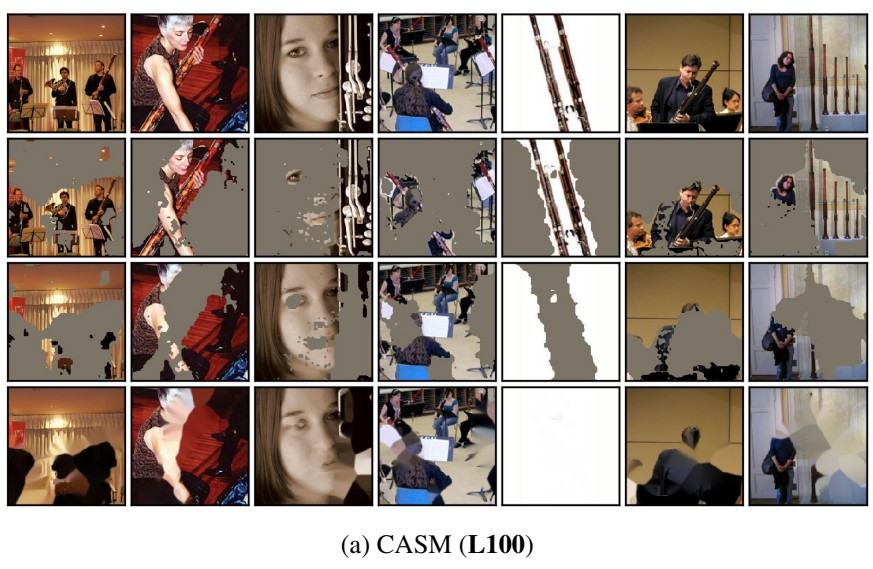

(a) CASM (**L100**)

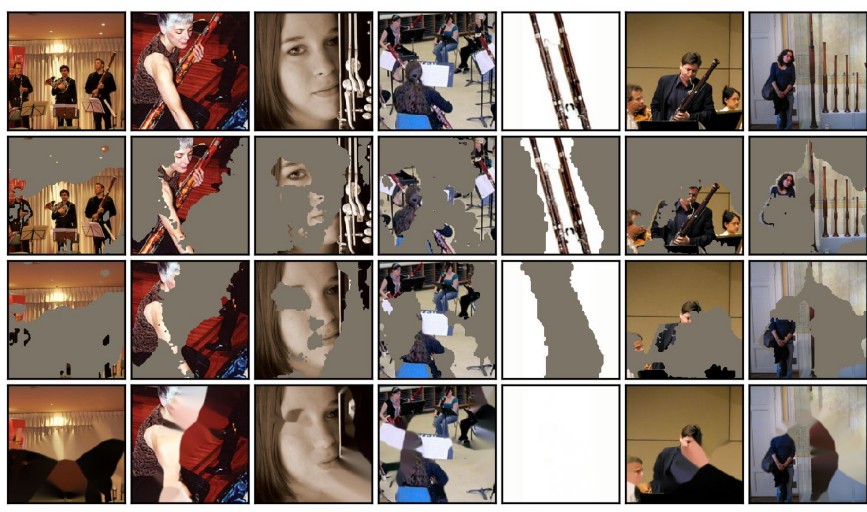

(b) CASM (**L**)

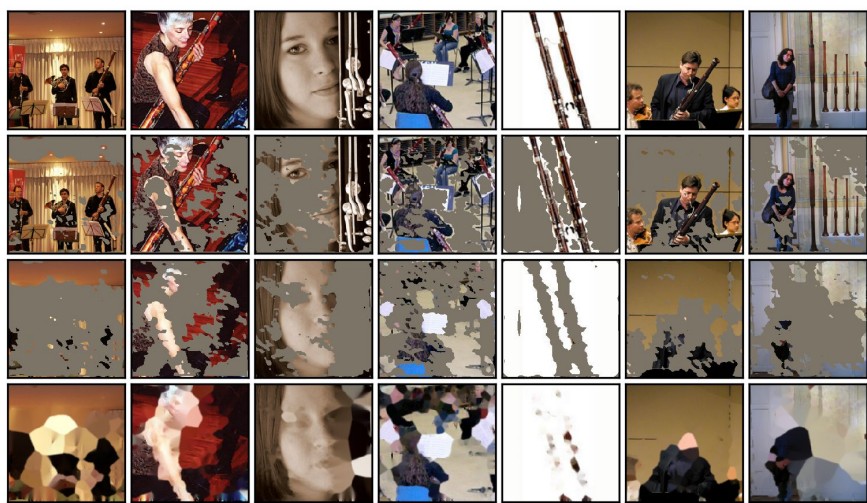

(c) Baseline

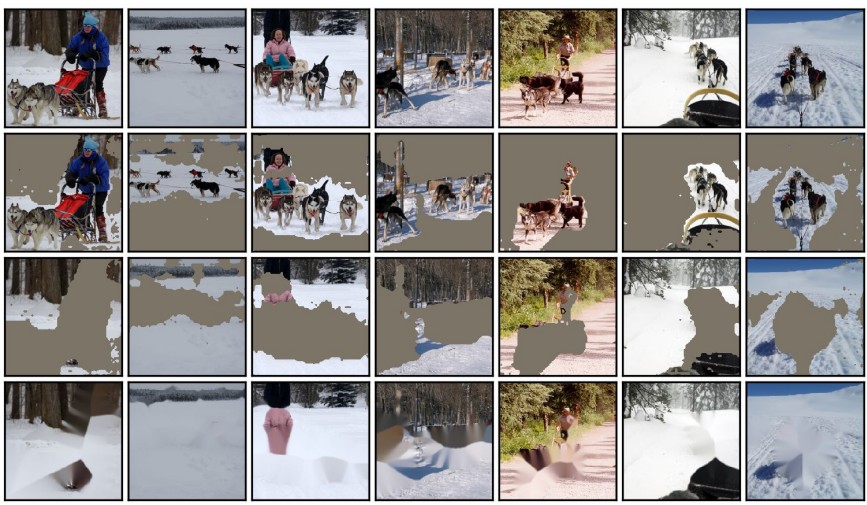

(a) CASM (**L100**)

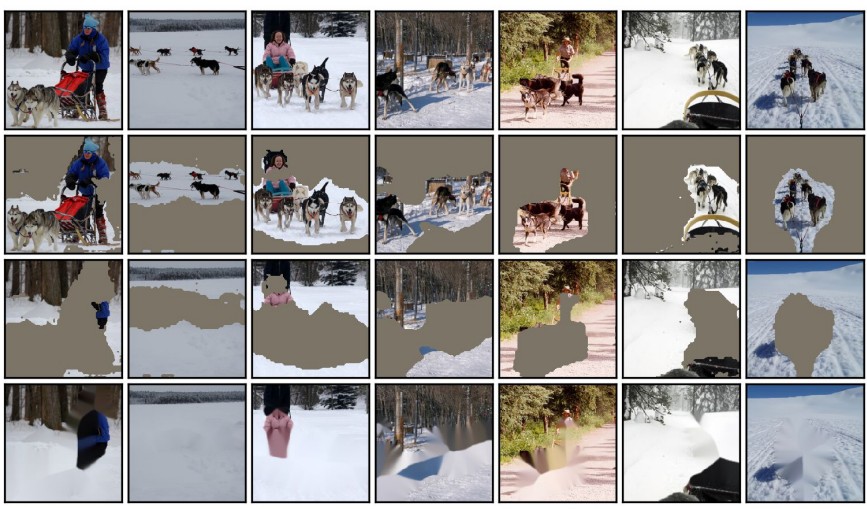

(b) CASM (**L**)

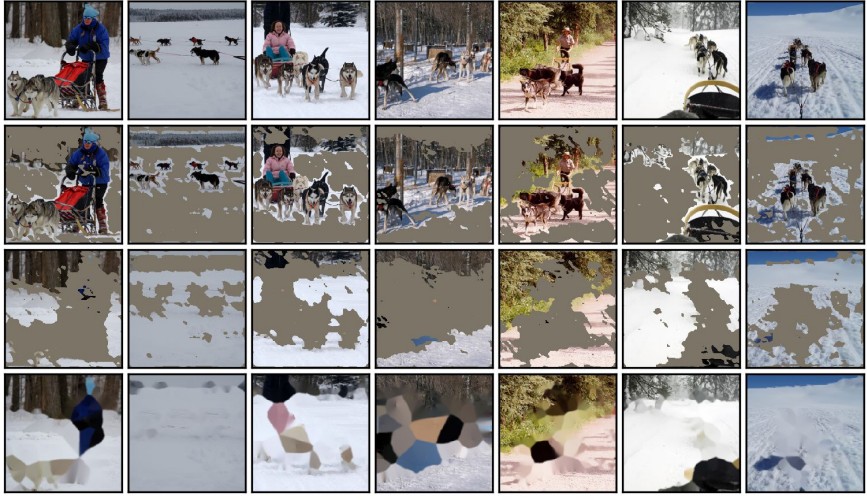

(c) Baseline

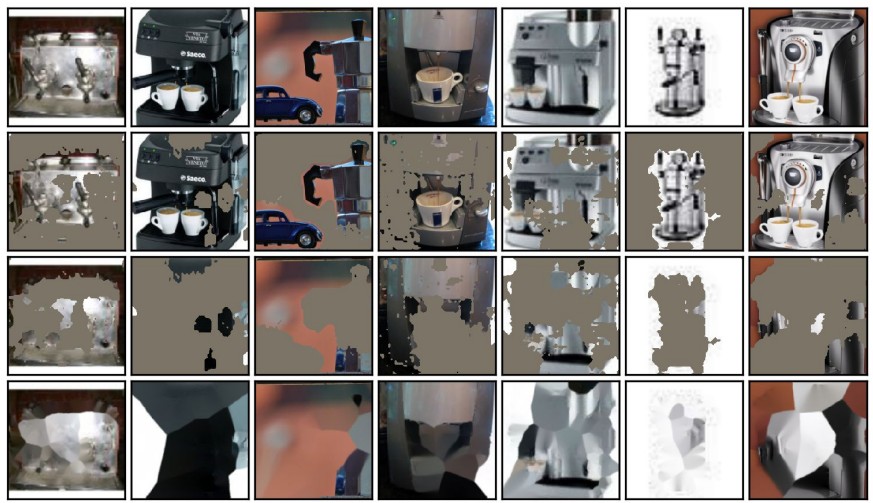

(a) CASM (**L100**)

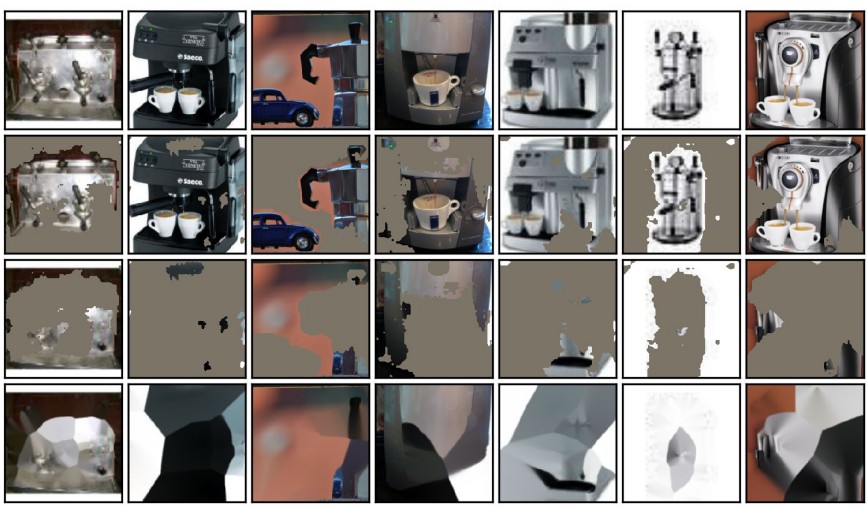

(b) CASM (**L**)

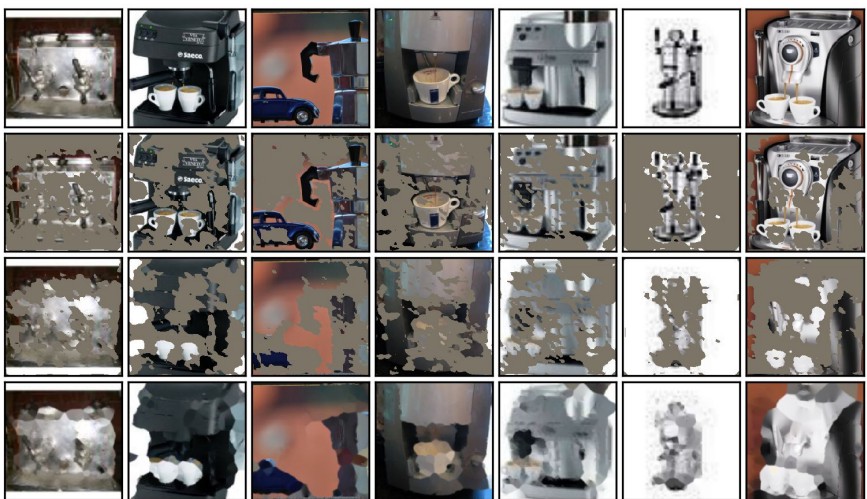

(c) Baseline

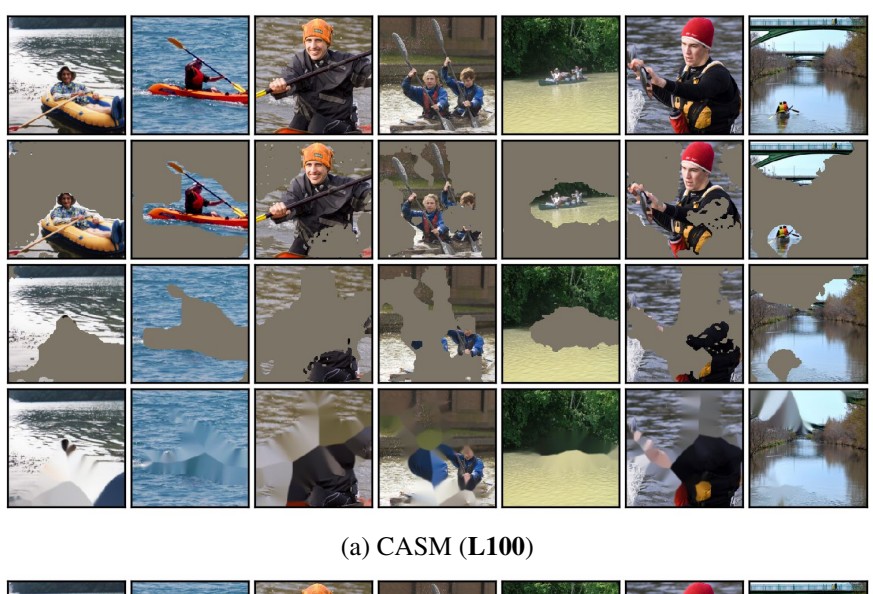

(a) CASM (**L100**)

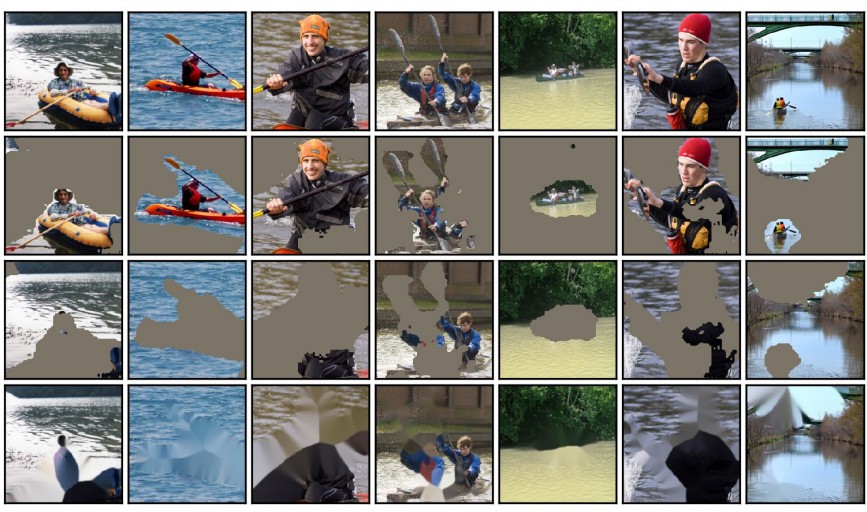

(b) CASM (**L**)

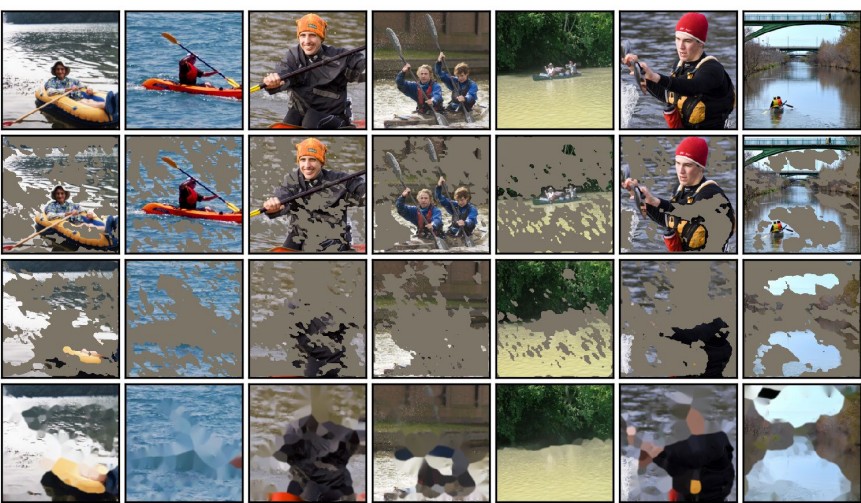

(c) Baseline

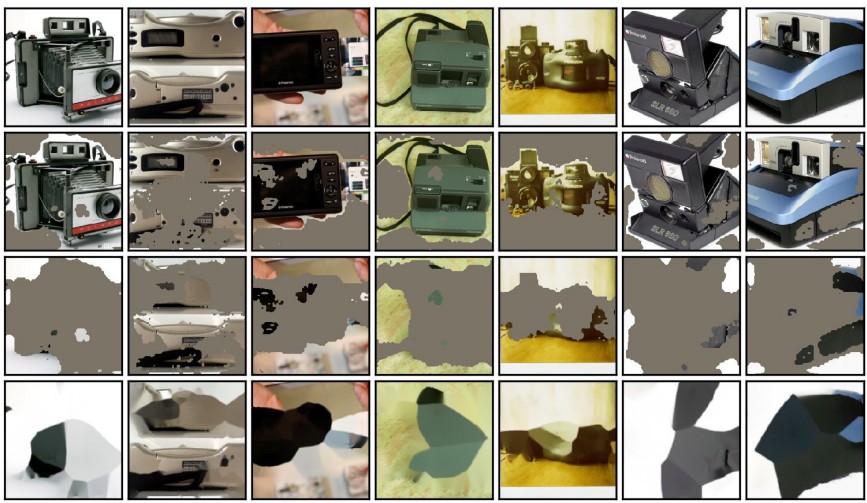

(a) CASM (**L100**)

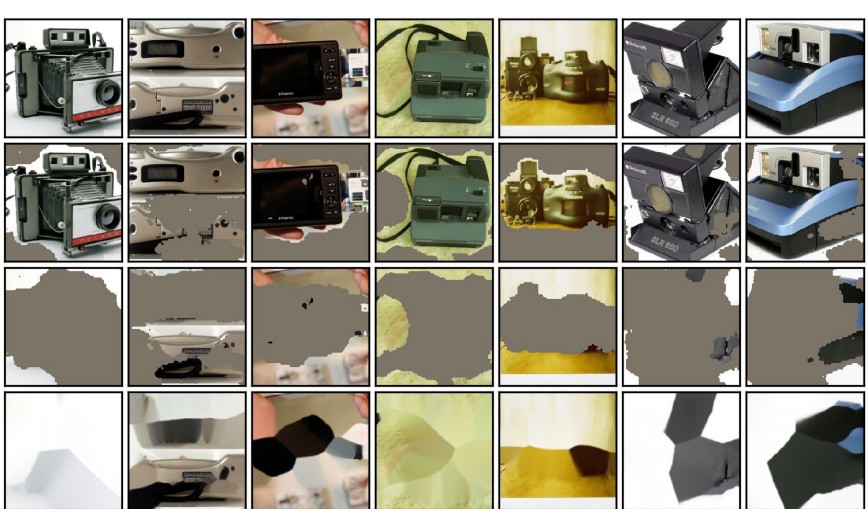

(b) CASM (**L**)

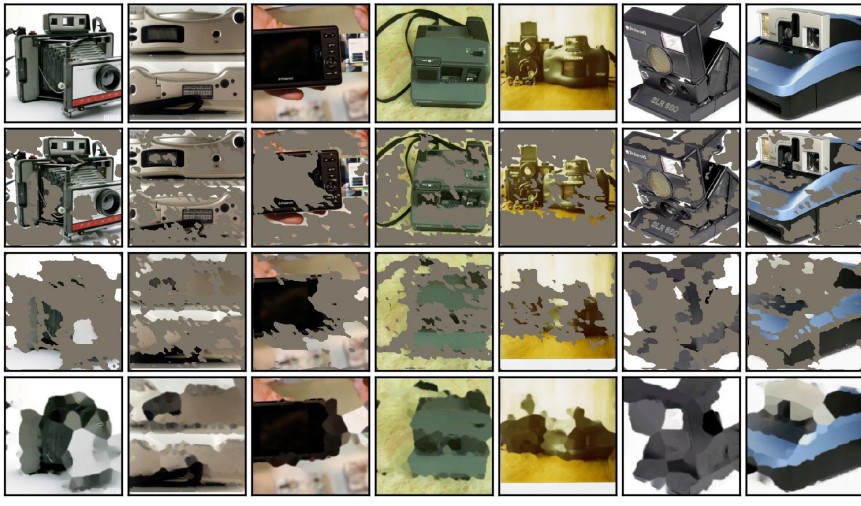

(c) Baseline

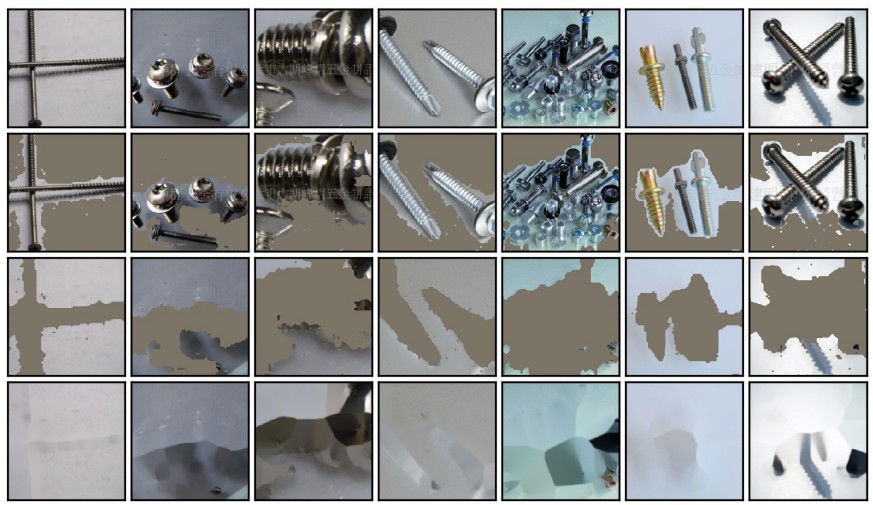

(a) CASM (**L100**)

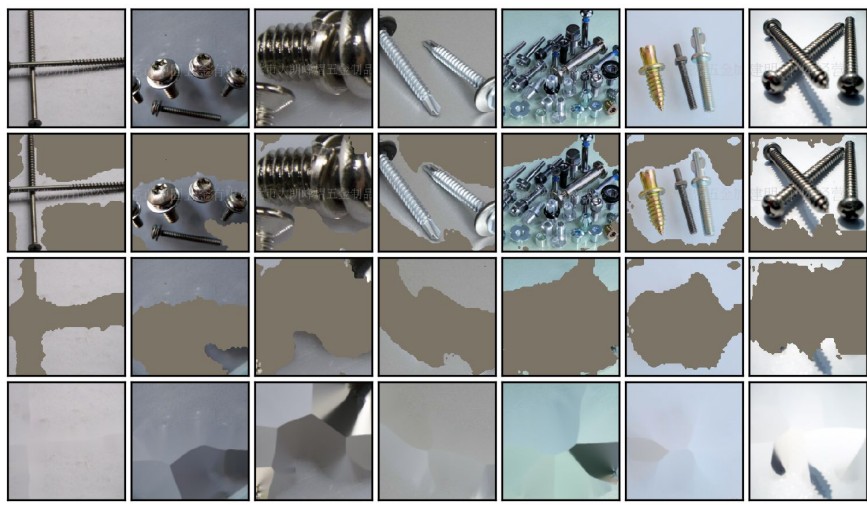

(b) CASM (**L**)

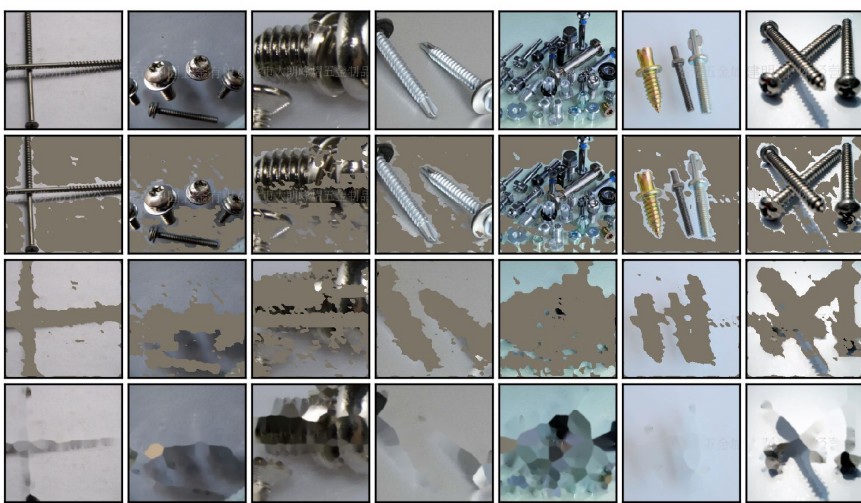

(c) Baseline

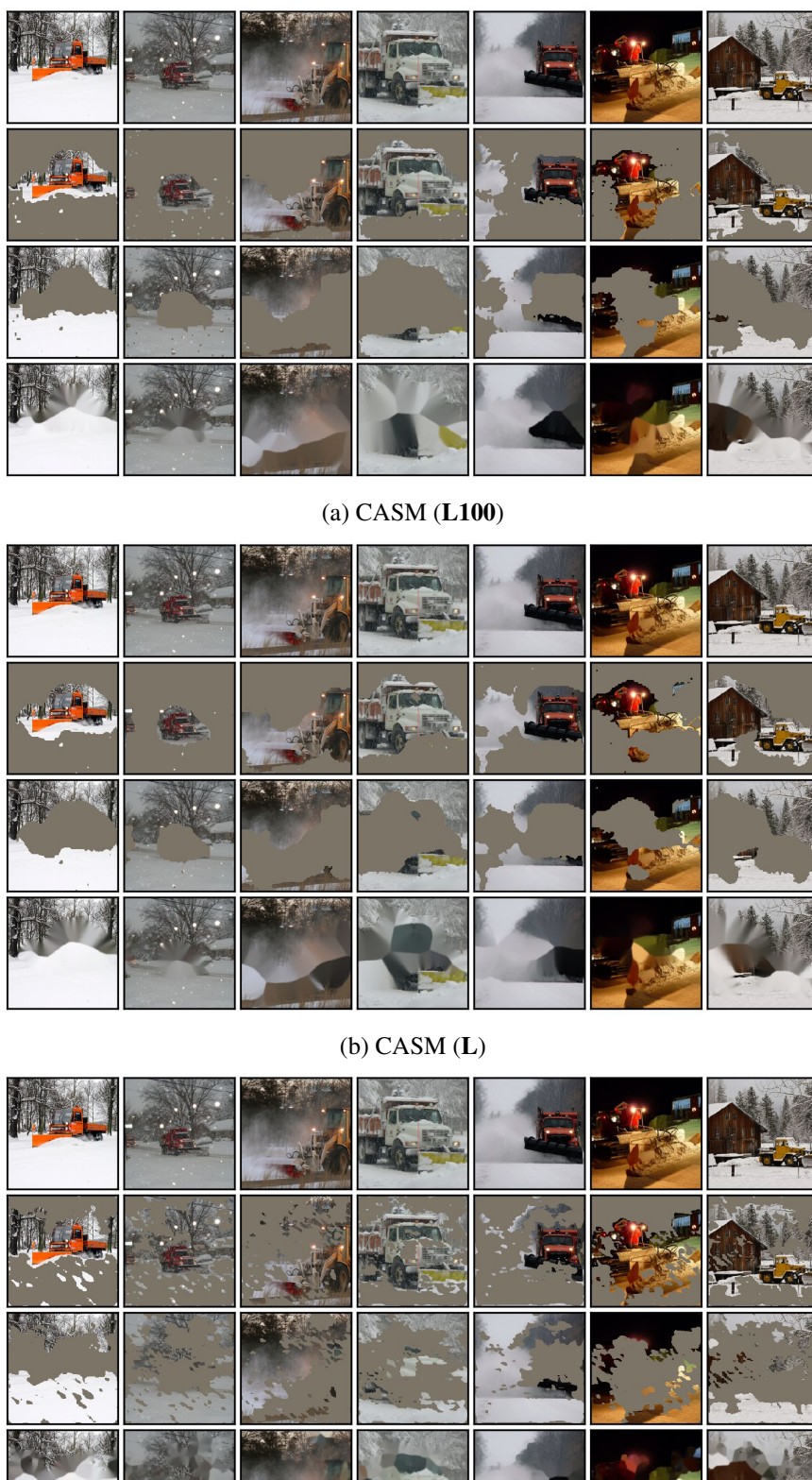

(a) CASM (**L100**)

(b) CASM (**L**)

(c) Baseline

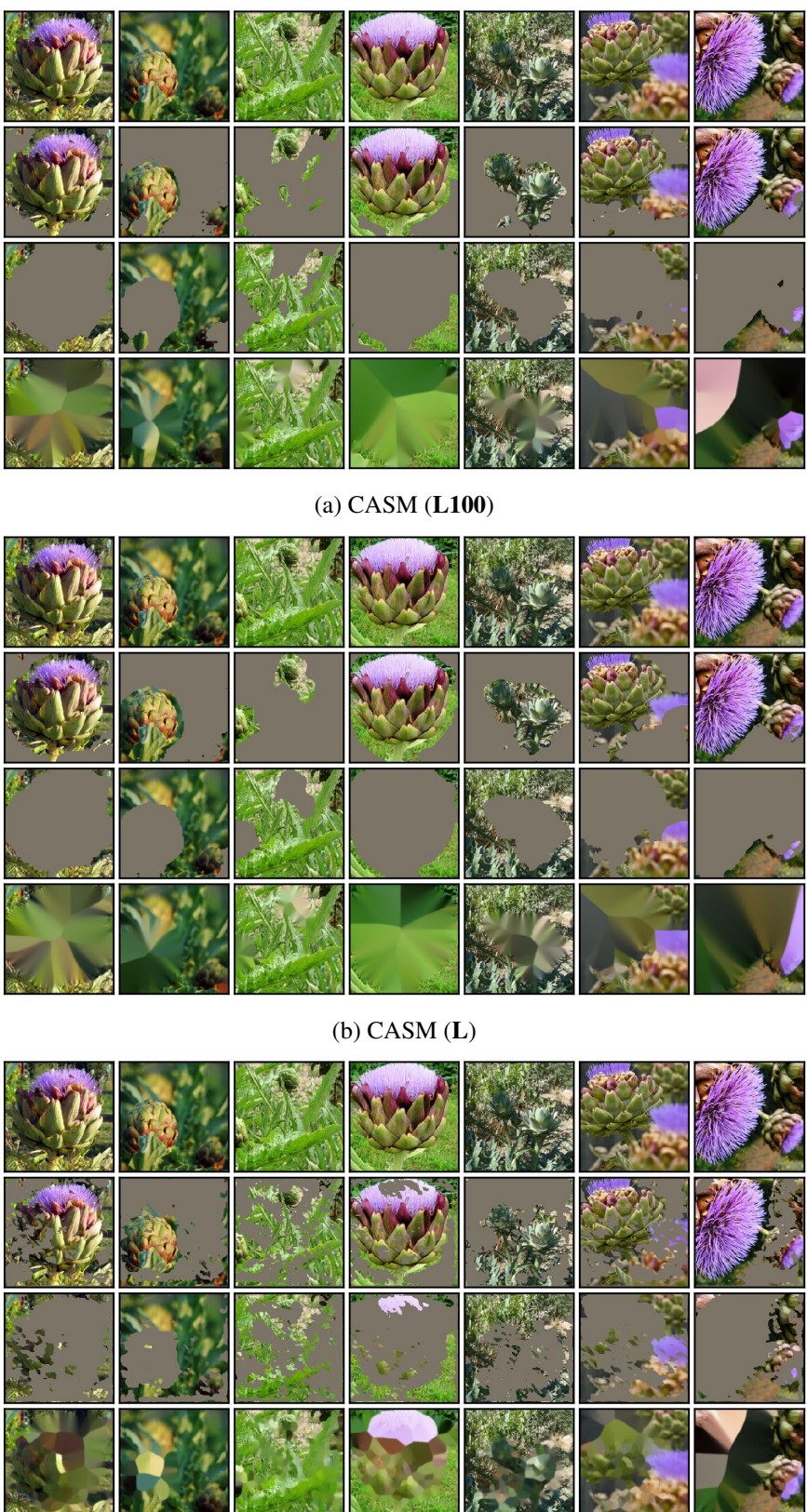

(a) CASM (**L100**)

(b) CASM (**L**)

(c) Baseline

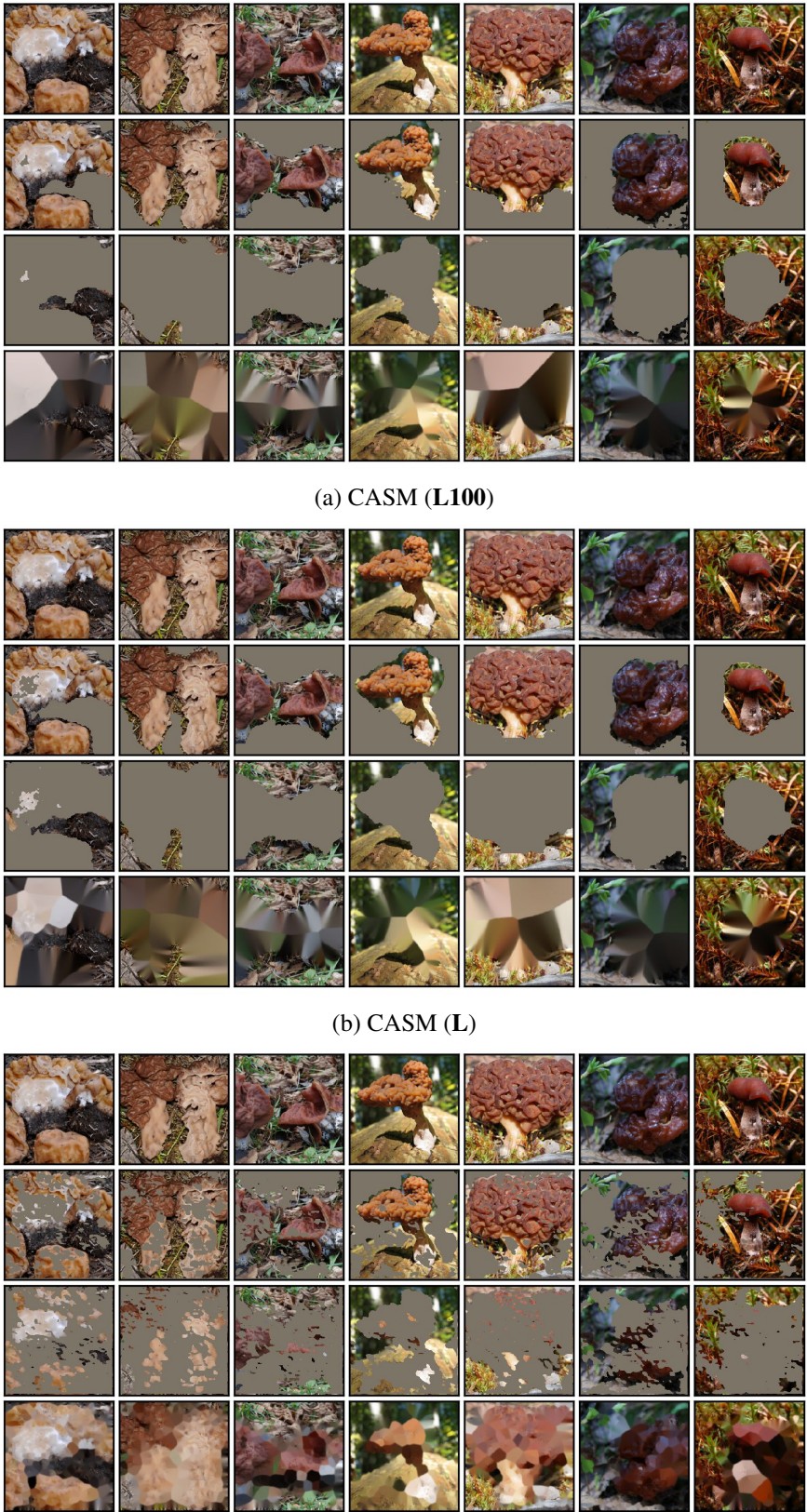

(a) CASM (**L100**)

(b) CASM (**L**)

(c) Baseline

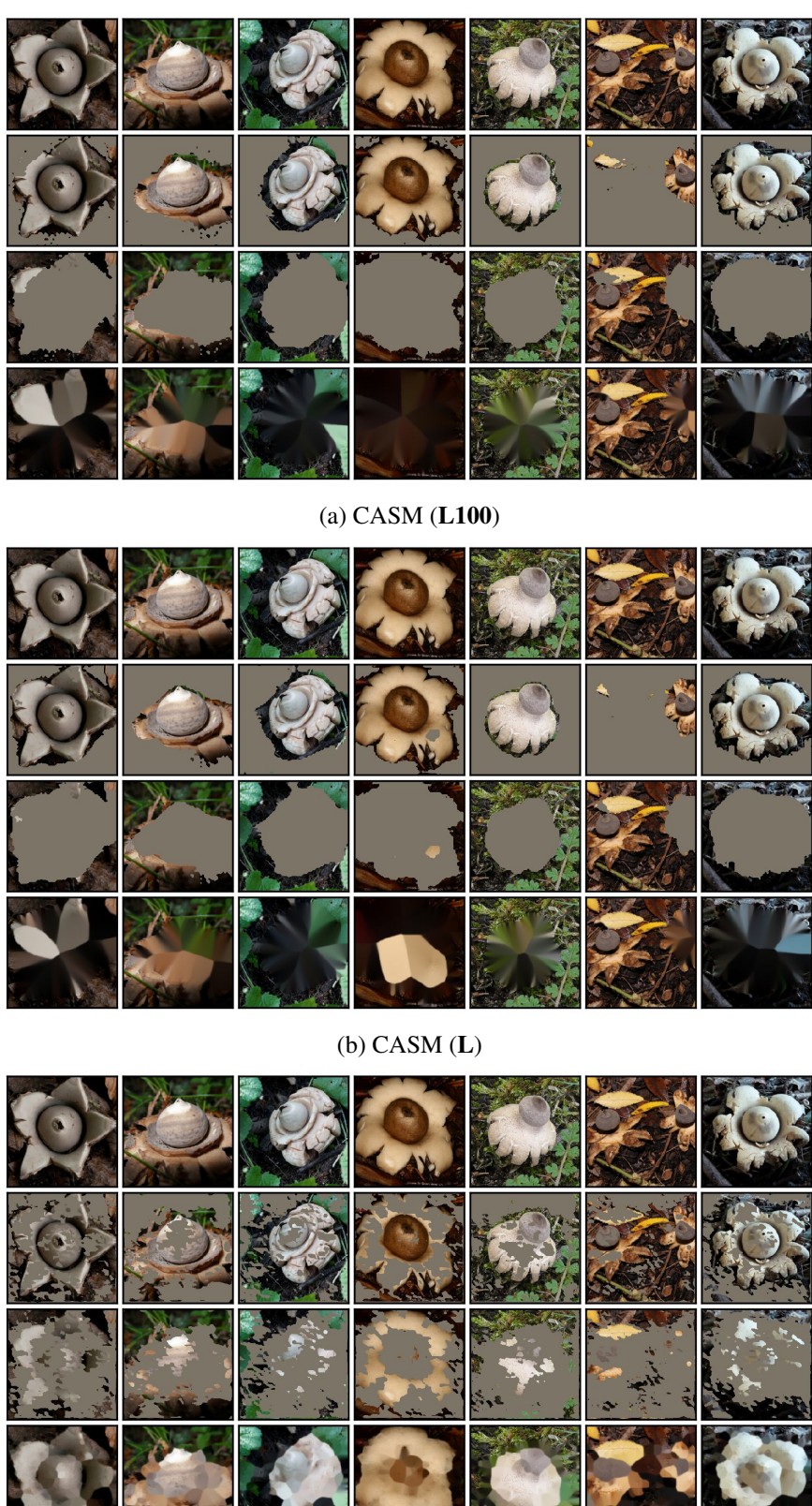

(a) CASM (**L100**)

(b) CASM (**L**)

(c) Baseline

