# OpenReview forum: "Classifier-agnostic saliency map extraction"
_ICLR.cc/2019/Conference_

### Official Review · AnonReviewer1 · 2018-11-03
**Issues with novelty and improvements in relation to prior works**

**Rating:** 4
**Confidence:** 4

**Review:**

This paper focuses on the extraction of high-quality model-agnostic saliency maps. The authors argue that when an extracted saliency map is directly dependent on a model, then it might not be useful for a different classifier and thus not general enough. To overcome this problem, they consider all the possible classifiers weighted by their posterior probabilities. This problem cannot be solved explicitly, and the authors suggest a scheme to approximate the solution using two networks. That is, pretrain an initial classifier and then, following an adversarial training procedure, one network is trying to confuse the classifier and the other one to maximize its accuracy. Using this formulation, the authors report state-of-the-art results for salience map extraction.

SUMMARY/OVERALL COMMENTS
The authors present a simple and effective way to produce classifier-agnostic saliency maps. The argument for the approach is well justified and the results seem convincing on a first read. However, the novelty of the method is a concern given the previous work of Fan et al. (2017), and the manuscript is not upfront about the differences between the two works. The experiments are another cause for concern: Fan et al. should have been tested as a baseline with similar implementation (controlling for architecture and \lambda), and implementation differences in prior works of Table 1 make it difficult to draw conclusions.


RELATED WORKS
* In the introduction, the authors mention related works but fail to mention the work of Fan et al. (2017) which is clearly the most relevant. The first mention of Fan et al is on page 4 in a very specific discussion the regularization coefficient. The problem formulation in Section 2 and the approach is Section 3 is largely borrowed from Fan et al but not acknowledged until the last page. This introduces bias and confusion to the reader in regards to the novelty of the approach. Please, mention the work of Fan et al. (2017) in the introduction and clearly delineate the differences in the works earlier in the text. (--)

* Du et al. (2018), “Towards Explanation of DNN-based Prediction with Guided Feature Inversion”, use the VGG models for saliency map extraction and achieve a LE of 38.2. Note that Du et al. (2018), suggest that this modification could lead to SOTA results. I would like to see a comparison with this method. (-)

* The work of Kindermans, et al. (2017), “Learning how to explain neural networks: PatternNet and Pattern Attribution”, although they do not aim for weakly supervised localization and thus, they do not present the LE, they produce saliency maps. I would like to see a LE comparison with that method. (minor -)

* In the introduction, p1 (last paragraph) other methods are briefly mentioned (Extracted saliency maps show all the evidence….superpixels), etc.) without references. Please add references when needed. (-)

* The framework presented in this paper was first proposed by Fan et al. (2017). The authors claim four main differences in their approach. In my eyes, not all of them are major or novel - probably the most impactful is removing superpixels as it simplifies the problem and implementation. (+)


APPROACH
* The authors aim for simplicity (strong +)

* The authors justify their approach and present their arguments clearly (strong ++)

* In the algorithm section, the authors first mention the sampling procedure and then their motivation. Please alter the ordering of these to be conceptually easier to understand your approach

* In the first sentence after the equation 6 I guess that “(cf. Alg. 1)” is a typo and should be modified to (Alg. 1)”

* After Equation 6 it is argued that the method resembles the training procedure of GANs (Godfellow et al., 2014) but not the work of Fan et al., 2017. (--)


EXPERIMENTS
* The authors define as their baseline the F thinning strategy (i.e. use only the first classifier) which is a model dependent salience map. While this is a useful comparison against the classifier dependent methods, given the similarity to the work of Fan et al. (2017), experiments comparing the proposed model to Fan et al. are necessary. It is important to control for network architecture (ResNet-50) and choice of \lambda to properly determine if the four changes outlined in Section 6 result in any real improvement over Fan et al. (strong --)

* The authors use the Table 1 (borrowed from Fong and Vedaldi (2017)) to compare their results against other methods. This comparison is problematic as different approaches are using different models as classifiers which may lead to increase or decrease of the LE. (--)

* In Table 2 the authors do not report how many times they run the same experiments to get these values. They also run less experiments with non-shared weights and they report only the LE. In my eyes it looks that the authors are trying to force their argument that the sharing weights helps (probably because it is one of their novelties). Please report the statistics of your experiments and fill the empty entries in the table. (--)

* In Table 3, what does the last row represent?

* Table 1 errors: (1) You write “Localization evaluation using OM, LE and F1 scores”. Please remove the F1 score as you do not report it. Also, correct the first sentence of the “Localization” subsection which states that you use three different metrics to “two different metrics”. (2) The LE from Fong and Vedaldi (2017) should be 43.2 and not 43.1.

* Regarding the unseen classes (section 5): (1) Please report in the appendix the classes that you are using in each subset. Are there classes correlated? (-)  (2) I see that there is a strong correlation between the LE on subset A and E. It looks like you are training on E and you generalize on A.


NOVELTY/IMPACT
* Novelty is a strong concern, given the work of Fan et al. (2017) (strong --). Nevertheless, the authors propose some changes that can be seen as more general, but the effectiveness of the changes is clearly established.

* This paper’s strongest point is the simplicity (conceptually and implementation-wise) of the method, an advantage over previous works (+)


OTHER COMMENTS
* Fan et al. (2017), use an adaptive λ that pushes the mask to 10% of the image whereas you are using a fixed one that pushes the mask to approximately 50% of the image. How can you make sure that this is not the reason that you are getting better results?

If the authors can clearly and fairly demonstrate that the changes they propose over Fan et al (2017) result in improved performance, and the manuscript is adjusted to be more upfront about this prior work, I would consider increasing my rating.

---

> ### Author Response · Authors · 2018-11-24
> **Reply**
>
> We thank you for the constructive comments.
>
> We fully agree that Fan et al. (2017) is the most relevant and very interesting work and it is why we devoted a full paragraph in our related work section to list all important differences. We thought that it will be much easier to understand the differences between these works when they are listed in the end of the paper and this is why we located the section there.
> We also agree that the comparison with previous works is problematic as different approaches are using different models as classifiers which influences the LE. We assumed however, and we believe it is a standard practice, that authors of the previous works did their best to achieve the best results. We also used relatively simple classifier (ResNet-50) so our results can be comparable with previous works. The same classifier was used by the previous SOTA work (Dabkowski et al. 2017) that further justifies our choice - we wanted to be comparable primarily with that work.
>
> Improvement over Fan et al. (2017) work is very large and we do not believe that tuning their method will remove the gap. Their approach is much more complex than ours (they use superpixels, gumbel trick, and heuristic adaptive strategy to fix regularization weight), which may also be a reason why their method is hard to tune and finally do not perform as well as ours. However, we do understand that their method was the first step in the right direction and we will try to make sure that we show enough respect for the work.
>
> We mentioned in the paper that the difference between our method and the previous SOTA is statistically significant. For ten separate training runs with random initialization the worst scores 36.3 and the best 36.0 with the average of 36.1 (that is reported in the paper). It means that the variance between the results is relatively small. Due to the limited computational resources we were not able to run ten trials for all ablation experiments conducted but we believe that the variance of results should be similar for all our experiments.
>
> We do not report OM score for nonshared architecture because in that case the model does not produce meaningful class predictions (it is not trained to do so). Hence, we will have to apply another classifier to get the OM score. However, the OM is always lower bounded by LE (and equals LE when an oracle classifier is used). As we can see in Table 2, even when the oracle was used, the results without sharing would be comparable to the results with a shared architecture using an ordinary ResNet-50 architecture (that achieves accuracy around 80%). We argue that the gap in LE is big enough to prove that sharing the weights is important.

---

> > ### Comment · AnonReviewer1 · 2018-12-04
> > **response to author feedback**
> >
> > The response and revised draft failed to address the following concerns:
> > * ​​The work of Fan et. al (2017) is still missing from the introduction while they report other related works as examples
> > * In the introduction, p1 (last paragraph) other methods are briefly mentioned  but still without references.
> > * A fair comparison with the work of Fan et. al (2017) is still missing (model architecture, fine-tunning etc.)
> > * In the ablation experiments, one run is not enough to evaluate the approach
> > * The correlation between the classes is not reported
> >
> > In addition, other minor details regarding the manuscript were not addressed.

---

> > > ### Author Response · Authors · 2018-12-09
> > > **a comment on reviewer's comments**
> > >
> > > We believe that the place where we cited Fan et al. (2017) is the right one, although we appreciate your feedback and will try to find a better place to position its appearance. While we appreciate this work, this work is not a major motivation behind our work. Unfortunately, its implementation is not available online, and its results are significantly lower than ours. Our method, albeit being much simpler, did not require any careful tuning of hyperparameters to achieve the accuracy we report in our submission, which we believe is a great advantage.

---

### Official Review · AnonReviewer2 · 2018-11-04
**Improvement (maybe) of saliency maps by introducing technical improvements over previous work**

**Rating:** 5
**Confidence:** 4

**Review:**

This paper introduces a new saliency map extractor that seems to improve state-of-the-art results. Saliency maps are tools that can be useful to understand the decision-making of deep networks for object recognition; advances in this research topic may lead to a better understanding of the functioning of deep networks.

The paper is based on the approach of Fan et al. (2017). The improvements over Fan et al. seem to be mainly technical: the objective function and the optimization procedure. This makes the novelty of the paper quite thin, as the main underlying idea of the paper was previously introduced.

The algorithm is introduced without motivating well the different choices. What are the intuitions and evidence that guided the design of the algorithm? How are the technical choices made? Answers to these questions may help to understand how this paper builds on Fan et al. and other previous works.

The experiments compare a comprehensive set of algorithms and show an improvement over previous works. Yet, the OM metric is only compared for a few of these algorithms and it is unclear why is so. Also, the qualitative examples do not clarify how the proposed method improves over state-of-the-art (it would be useful to compare with qualitative examples of previous work). It remains unclear how much of an improvement over state-of-the-art there is.

In summary, I think the paper could be valuable as the proposed algorithm may improve state-of-the-art results. Yet, these results and the novelty of the paper are not entirely clear.

---

> ### Author Response · Authors · 2018-11-24
> **Reply**
>
> We thank the reviewer for the detailed comments.
>
> We tried to list all important changes between our work and ALN (Fan et al. 2017) and we devoted a full paragraph in our related work section to achieve that. There are four major differences between that work and ours. First, we use the entropy as a score function for training the mapping, whereas they used the classification loss. This results in obtaining better saliency maps as we have shown earlier. Second, we make the training procedure faster thanks to tying the weights of the encoder and the classifier, which also results in a much better performance. Third, we do not let the classifier shift to the distribution of masked-out images by continuing training it on both clean and masked-out images. Finally, their mapping relies on superpixels to build more contiguous masks which may miss small details due to inaccurate segmentation and makes the entire procedure more complex. Our approach solely works on raw pixels without requiring any extra tricks. We believe that these differences are not just technical improvements and we provided a set of ablation studies to present how the particular choices were made.
>
> The official metric (OM) from The ImageNet Large Scale Visual Recognition Challenge (ILSVRC)  considers the localization successful if at least one ground truth bounding box has IOU with predicted bounding box higher than 0.5 and the class prediction is correct. Since OM is dependent on the classifier, from which we have sought to make our mapping independent, we use another widely used metric, called localization error (LE), which only depends on the bounding box prediction. We believe that dependence on the classifier is the main reason why other works don't report OM metric. However, since OM was the official metric it should be reported and it is why we computed that for our models. We reported the value for pior works that provide the value.

---

> > ### Comment · AnonReviewer2 · 2018-12-14
> > **Reply**
> >
> > Thanks for the clarifications.
> >
> > I am left with the impression that the improvements on Fan et al. are more like "tricks" than something fundamental that next algorithms will build on.
> >
> > Also, I would indicate in the paper that the results from previous works are not reproduced but the numbers are copied from the papers.

---

### Official Review · AnonReviewer3 · 2018-11-04
**classifier-agnostic method for object localization**

**Rating:** 4
**Confidence:** 3

**Review:**

This paper proposes a classifier-agnostic method for saliency map extraction. In order to address the dependence of saliency map extraction on the classifier, the authors propose to learn a saliency mapping by considering all possible classifiers (i.e., a certain classifier structure w.r.t. the space of all its parameters). The goal is to find the relevant features in the data that work with all possible classifiers. The proposed framework is formulated as a min-max game between two players: a mask m corresponding to the saliency mapping, and a function f sampled from a set of classifiers with the same structure but different parameters. The mapping m is optimized to maximize the masked-out classification error (such that m captures all relevant features whose removal can maximally confuse the classifier), while f is optimized to minimize the mask-out classification error.

The idea of how to formulate the classifier set and how to sample from the set is interesting. However, I have some concerns regarding the overall model:

1) It seems not quite convincing to me why the model should involve an adversarial game. In particular, why f should be optimized to minimize the masked-out classification error? I understand that by doing this, f has an opposite goal with m so as to force m to capture as many relevant features as possible. However, I do not think f has a natural motivation to minimize the maxed-out error. In my opinion, it seems more convincing if f is optimized to minimize the masked-in classification error, but not necessarily the masked-out one. I think this also explains why the model works better by adding the classification loss over original images in Eq. (8). Would it be more natural if we optimize both f and m to minimize the masked-in classification error? And it would easier to train compared with the min-max model. Maybe some more explanation on the motivation of such an adversarial game can be helpful.

2) I was curious whether the alternating optimization of m and f would cause the cumulation of errors? I mean, if in some iteration m just captures irrelevant features, f will still be optimized to accomodate to such a bad mapping. Would such kind of error accumulate during training?

3) In Algorithm 1, after \theta_m is learned, how is m is determined?

---

> ### Author Response · Authors · 2018-11-24
> **Reply**
>
> Thanks for the review and the feedback.
>
> We believe that adversarial game, as noted in the review, forces a mapping m to capture as many relevant features as possible and it is the main objective of our work. The approach that you suggest (minimizing the masked-in classification error) is also interesting but it will end up in having a mapping m that highlights the most important part for classifier that is trained with, hence it will not be classifier-agnostic. It will also not provide an incentive for the classifier change (the same features can be still used), which makes the idea of using previous iteration of the classifier not useful.
>
> The phenomenon that you mentioned in 2) is not a problem when adversarial game is used. If the mapping is bad (i.e. does not cover important features), the classifier f will easily predict the correct class. As a result, the optimization procedure will not change the classifier f significantly.
>
> \theta_m are the weights of the mapping m. Hence, the relation between them is straightforward. We will make that clear in the next version of the paper.

---

### Meta-Review · Area_Chair1 · 2018-12-13
**interesting problem and method; insufficient discussion of and comparison to related work**

**Confidence:** 3
**Recommendation:** Reject

**Metareview:**

{418}; {Classifier-agnostic saliency map extraction}; {Avg: 4.33}; {}

1. Describe the strengths of the paper.  As pointed out by the reviewers and based on your expert opinion.

The paper is well-written and the method is simple, effective, and well-justified.

2. Describe the weaknesses of the paper. As pointed out by the reviewers and based on your expert opinion. Be sure to indicate which weaknesses are seen as salient for the decision (i.e., potential critical flaws), as opposed to weaknesses that the authors can likely fix in a revision.

1. The introduction, in particular the last row of pg 1, implies that this work is the first to show that a class-agnostic saliency estimation method can produce higher-quality saliency maps than class-dependent ones. However, Fan et al. have already shown this. For this reason, AR1 recommended that the authors reword the introduction to reflect prior work on this aspect but the authors declined to do so. The AC would have liked to see a discussion of how the different points of view of the two works (robustness to corruption vs class-agnosticism) both address the same issue (poor segmentation of the salient image regions).
2. The work of Fan et al has a very similar approach and a deeper comparison is needed. While the authors dedicated two paragraphs of discussion to this work, they should have gone further. For example, the work of Fan et al. uses a very simple saliency map extraction network and it's unclear how much this impacts their performance when compared to the proposed method, which uses ResNet50. The AC agrees with the authors that re-implementing the method of Fan et al. is asking a lot but a discussion of the potential impact would have sufficed.
3. The authors didn't mention at all the vast body of work on salient object detection (for a somewhat recent review see Borji et al. "Salient object detection: A benchmark." IEEE TIP). The differences to this line of work should have been discussed.

Points 1 and 2 were particularly salient for the final decision.

3. Discuss any major points of contention. As raised by the authors or reviewers in the discussion, and how these might have influenced the decision. If the authors provide a rebuttal to a potential reviewer concern, it’s a good idea to acknowledge this and note whether it influenced the final decision or not. This makes sure that author responses are addressed adequately.

Two major points of contention were:
- The discussion of differences between the proposed method and the method of Fan et al.
- The fairness of the comparison to Fan et al.
AR1 felt that the paper was deficient on both counts (AR2 had similar concerns) and the authors disagreed, arguing that the discussion was complete and the quantitative comparison fair.

The AC was sympathetic to these concerns and found the authors' responses to be dismissive of those concerns. In particular, the AC agrees that the paper, as currently organized, minimizes the degree to which the work is derived from Fan et al.

4. If consensus was reached, say so. Otherwise, explain what the source of reviewer disagreement was and why the decision on the paper aligns with one set of reviewers or another.

The reviewers reached a consensus that the paper should be rejected.